# Phytoplankton and Bacterial Community Structure in Two Chinese Lakes of Different Trophic Status

**DOI:** 10.3390/microorganisms7120621

**Published:** 2019-11-27

**Authors:** Cui Feng, Jingyi Jia, Chen Wang, Mengqi Han, Chenchen Dong, Bin Huo, Dapeng Li, Xiangjiang Liu

**Affiliations:** College of Fisheries, Hubei Province Engineering Laboratory for Pond Aquaculture, Huazhong Agricultural University, Wuhan 430070, China; fc047142@163.com (C.F.); jiajy94@163.com (J.J.); chenwangc163@163.com (C.W.); ahanmengqi@163.com (M.H.); dc15623555078@163.com (C.D.); huobin@mail.hzau.edu.cn (B.H.); ldp@mail.hzau.edu.cn (D.L.)

**Keywords:** bacterial community, phytoplankton, oligotrophic lake, eutrophic lake, 16S rRNA

## Abstract

Phytoplankton are the primary producers at the basis of aquatic food webs, and bacteria play an important role in energy flow and biochemical cycling in aquatic ecosystems. In this study, both the bacterial and phytoplankton communities were examined in the oligotrophic Lake Basomtso and the eutrophic Lake South (China). The results of this study showed that the phytoplankton density and diversity in the eutrophic lake were higher than those in the oligotrophic lake. Furthermore, *Chlorophyta* (68%) and *Cryptophyta* (24%) were the dominant groups in the eutrophic lake, while *Bacillariophyta* (95%) dominated in the oligotrophic lake. The bacterial communities in the waters and sediments of the two lakes were mainly composed of *Proteobacteria* (mean of 32%), *Actinobacteria* (mean of 25%), *Bacteroidetes* (mean of 12%), and *Chloroflexi* (mean of 6%). Comparative analysis showed that the abundance of bacteria in the eutrophic lake was higher than that in the oligotrophic lake (*p* < 0.05), but the bacterial diversity in the oligotrophic lake was higher than that in the eutrophic lake (*p* < 0.05). Finally, the bacterial abundance and diversity in the sediments of the two lakes were higher than those in the water samples (*p* < 0.05), and the *Latescibacteria* and *Nitrospinae* groups were identified only in the sediments. These results suggest that both the phytoplankton and bacterial communities differed considerably between the oligotrophic lake and the eutrophic lake.

## 1. Introduction

Microorganisms are both the producers and decomposers of organic matter in aquatic ecosystems and play an important role in regulating the circulation of biogenic elements such as carbon, nitrogen, phosphorus, and sulfur in lakes [1,2]. Bacteria, as an important part of the microbial community, are mainly responsible for the mineralization and recycling of organic matter, while the recycling of dissolved organic carbon (DOC) in lakes is mainly fulfilled by heterotrophic bacteria [3,4,5]. Phytoplankton are the primary producers at the basis of aquatic food webs and can quickly respond to environmental changes [6]. In lakes, interactions between phytoplankton and bacteria have been proposed to influence bacterial community dynamics [7,8,9]. Bacteria rapidly utilize exudates released by phytoplankton (e.g., sugars, amino acids), as well as detritus following algal cell death [7,10]. In addition to providing a source of organic matter, phytoplankton can provide a habitat for endophytic bacteria living within algae cells and epiphytic bacteria that live in the phycosphere surrounding algal cells [11,12]. Meanwhile, phytoplankton could also display a negative effect on the bacterial community through nutrient competition and antibiotic release [13]. 

According to biological productivity, lakes can be divided into three trophic types: oligotrophic, mesotrophic, and eutrophic lakes [14,15]. In an aquatic environment, the composition and diversity of the bacterial community may vary with the water quality. Generally, the community dynamics of aquatic bacteria vary with biotic and abiotic environmental variables, e.g., temperature, availability of nutrients, predation, and interactions with other organisms, including phyto- and zooplankton [16,17,18]. Previous studies have showed that the main driving factors, including nitrogen, phosphorus, and temperature, could alter the taxonomical structure of the bacterial community in freshwater lakes [19,20,21]. However, these studies were mainly focused on the taxonomical composition of the sediment community [22,23,24,25]. Little is known about the bacterial community in different trophic waterbodies [26]. Comparing the structures of bacterial communities in different trophic lakes could provide valuable information to protect and remediate these lakes. Hence, it is essential to explore the bacterial communities in different trophic lakes.

Previous studies mainly focused on the bacterial diversity in lake sediments using traditional isolation methods [27,28] and conventional DNA-based molecular methods (e.g., DGGE, T-RFLP, Q-PCR, FISH, RAPD, clone libraries) [29,30,31,32,33,34]. Recently, 16S rRNA sequencing on the Illumina MiSeq platform has been able to provide more detailed information about microbial community diversity and structure [35,36,37,38]. This technique has been widely used in the early detection of aquatic invasion and in investigations of water biodiversity [38]. It can not only judge the distribution of species and analyze the community structure of species but also effectively improve the efficiency and quality of aquatic ecosystem detection. Furthermore, it could be used to formulate corresponding effective protection measures for maintaining ecological balance [38,39,40]. 

In this study, to clarify the phytoplankton and bacterial community structures in two different trophic lakes, the oligotrophic Lake Basomtso and eutrophic Lake South were used as models. Firstly, the phytoplankton community composition in water samples from the two lakes was examined. Secondly, the bacterial community in the water and sediment from the two lakes was analyzed by using 16S rRNA sequences. Finally, to determine whether bacteria showed different assembly patterns in different habitats, the present study also compared the bacterial community composition in the water and sediment of the oligotrophic lake.

## 2. Materials and Methods

### 2.1. Study Sites and Sample Collection

The study was conducted in oligotrophic Lake Basomtso and eutrophic Lake South. Lake Basomtso (area of 27 km^2^, maximum depth of 120 m, and mean depth of 68 m) is one of the largest dammed freshwater lakes in the southeast of Tibet. Lake South (area of 5.50 km^2^, maximum depth of 3.2 m, and mean depth of 1.6 m) is located in Wuhan City in the Middle-Lower Yangtze plains. From both lakes, samples were collected in August 2018. At oligotrophic Lake Basomtso, samples were collected from the surface (O-S1, O-S2, O-S3) and bottom water layers (O-B1, O-B2), as well as from the sediments (O-SE1, O-SE2, O-SE3) from three sites situated from the river inlet to the lake center. At eutrophic Lake South, samples were collected from the surface water layers (E-S1, E-S2, E-S3) and from the sediments (E-SE1, E-SE2) from three sites situated from the shore to the center. The locations of the studied lakes and sampling sites are presented in Figure 1. 

At each site, the water temperature, pH, and dissolved oxygen (DO) were measured in situ using a HACH portable multi-parameter meter (HACH, HQ40d, Loveland, CO, USA). Surface water samples were collected from about 1 m under the water surface, and the bottom water samples were collected from about 1–2 m above the sediment. Water samples were collected using a 5 L plexiglass water collector and poured into 5 L brown plastic bottles. Sediment samples were collected using an HL-CN mud collector and transferred to polypropylene sealed bags. All the samples were stored in an incubator at 4 °C and transported to the laboratory within two hours. 

For molecular analysis, 1000 mL water samples were prefiltered through a 20 μm membrane (Millipore, Carrigtwohill, Co, Cork, Ireland) and subsequently filtered onto 0.22 μm pore size polycarbonate filters (47 mm, Millipore). The filters were placed into 2 mL sterile tubes and immediately frozen in liquid nitrogen, then stored at −80 °C until further analysis. The sediment samples were also stored at −80 °C until further procedures.

### 2.2. Qualitative and Quantitative Analyses of Phytoplankton

Samples for the identification and enumeration of phytoplankton were collected using a 1 L sampler and fixed with Lugol’s solution (2% final concentration). In the laboratory, the samples were concentrated (20×) by sedimentation and preserved with 1 mL formaldehyde. Phytoplankton from the oligotrophic lake were transferred to a 10 mL Utermöhl Chamber (Hydro-bios, Kiel, Germany) and counted under an inverted microscope (Nikon Eclipse T*s*2, Kobe, Japan) at 400× magnification. Meanwhile, phytoplankton samples from the eutrophic lake were identified and counted with an optical microscope (Nikon Eclipse E100, Kobe, Japan) using a 0.1 mL counting chamber at 400× magnification. Besides this, species identification was performed as in previous studies [41].

### 2.3. DNA Extraction, PCR Amplification, and Illumina MiSeq Sequencing

Bacterial DNA was extracted from all water and sediment samples using the E.Z.N.A.^®^ soil DNA Kit (Omega Bio-tek, Norcross, GA, USA) according to the manufacturer’s protocols. The final DNA concentration and purification were determined using a NanoDrop 2000 UV–vis spectrophotometer (Thermo Scientific, Wilmington, NC, USA), and the DNA quality was checked via 1% agarose gel electrophoresis. The V3–V4 hypervariable regions of the bacteria 16S rRNA gene were amplified with primers 338F (5′-ACTCCTACGGGAGGCAGCAG-3′) and 806R (5′-GGACTACHVGGGTWTCTAAT-3′) using a thermocycler PCR system (GeneAmp 9700, ABI, Foster City, CA, USA). The PCR reactions were conducted using the following program: 3 min of denaturation at 95 °C; 27 cycles of 30 s at 95 °C, 30 s for annealing at 55 °C, and 45 s for elongation at 72 °C; and a final extension at 72 °C for 10 min. PCR reactions were performed in triplicate in a 20 μL mixture containing 4 μL of 5× FastPfu Buffer, 2 μL of 2.5 mM dNTPs, 0.8 μL of each primer (5 μM), 0.4 μL of FastPfu Polymerase, and 10 ng of template DNA. The PCR products were extracted from a 2% agarose gel, further purified using the AxyPrep DNA Gel Extraction Kit (Axygen Biosciences, Union City, CA, USA), and quantified using QuantiFluor™-ST (Promega, Madison, WI, USA) according to the manufacturer’s protocol. Purified amplicons were pooled in equimolar quantities and paired-end sequenced (2 × 300) on an Illumina MiSeq platform (Illumina, San Diego, CA, USA) according to the standard protocols by Majorbio Bio-Pharm Technology Co. Ltd. (Shanghai, China). The raw reads were deposited into the NCBI Sequence Read Archive (SRA) database (Accession Number: SRR9612299~SRR9612323 (25 objects)).

### 2.4. Data Analysis

Raw reads were demultiplexed, quality filtered by Trimmomatic, and merged by FLASH with the following criteria: (i) The reads were truncated at any site receiving an average quality score of <20 over a 50 bp sliding window. (ii) Primers were exactly matched allowing two nucleotide mismatches, and reads containing ambiguous bases were removed. (iii) Sequences with overlap longer than 10 bp were merged according to their overlap sequence. Operational taxonomic units (OTUs) were clustered with a 97% similarity cutoff using UPARSE (version 7.1 http://drive5.com/uparse/) and chimeric sequences were identified and removed using UCHIME. The taxonomy of each 16S rRNA gene sequence was analyzed using the RDP Classifier algorithm (http://rdp.cme.msu.edu/) against the Silva (SSU123) 16S rRNA database using a confidence threshold of 70%. Nonparametric indicators (Chao1, Shannon, and coverage) were used to evaluate the relationships among bacterial community diversity characteristics and community coverage between Lake Basomtso and Lake South. Principal co-ordinates analysis (PCoA) was employed to explore and visualize the similarities between the samples obtained from the two lakes based on Bray–Curtis dissimilarity using the package Ape (Version 5.3). The three replicates are expressed as the mean ± SD; all data were tested for normality of distribution using the Shapiro–Wilk normality test. Then, one-way ANOVA was used to test for significant differences according to the different experiment groups. The differences between groups were considered significant at * *p* < 0.05 and highly significant at ** *p* < 0.01 or at *** *p* < 0.001.

## 3. Results

### 3.1. Environmental Conditions

As shown in Table 1, both the water temperature and the pH value were significantly lower in oligotrophic Lake Basomtso than in eutrophic Lake South (*p* < 0.001). The DO was slightly lower in oligotrophic Lake Basomtso than in eutrophic Lake South. The total nitrogen (TN) and total phosphorus (TP) in oligotrophic Lake Basomtso were also significantly lower than those in eutrophic Lake South (*p* < 0.01). In addition, the chlorophyll *a* (Chl-*a*) level in eutrophic Lake South was significantly higher than that in oligotrophic Lake Basomtso (*p* < 0.001). These results indicated that there are significant differences in physicochemical characteristics between the eutrophic lake and the oligotrophic lake.

### 3.2. Phytoplankton Community Composition in the Two Lakes

The total phytoplankton density in eutrophic Lake South (4.82 × 10^8^ cells/L) was about 10,000 times higher than that in oligotrophic Lake Basomtso (3933 cells /L) (Table 1). *Bacillariophyta* accounted for 94.96% of the total phytoplankton density in the oligotrophic lake, whereas *Chlorophyta* and *Cyanophyta* accounted for 68.26% and 24.48%, respectively, of the total phytoplankton density in the eutrophic lake. Additionally, the number of phytoplankton species in the oligotrophic lake (21 genera, 6 phyla) was lower than that in the eutrophic lake (50 genera, 6 phyla) (Appendix A). The phytoplankton community in oligotrophic Lake Basomtso was mainly dominated by *Cyclotella meneghiniana*, *Nitzschia linearis*, and *Iconella biseriata* (Appendix A). In contrast, the main dominant species in eutrophic Lake South were *Dolichospermum circinale, Dolichospermum viguieri, Microcystis wesenbergii*, and *Merismopedia minima* (Appendix A).

### 3.3. Bacterial Community Structure in the Two Lakes

A total of 1,213,937 high-quality sequences was obtained from all the samples. These sequences were classified to define OTUs based on a similarity threshold of 97%. The rank abundance curve showed that all sequencing depths were sufficient to reflect the bacterial diversity (Figure 2A). Good’s coverage (≥99%) indicated a high degree of sequence coverage (Table 2). The nonparametric Shannon analysis showed that the bacterial diversity in the oligotrophic lake was higher than that in the eutrophic lake (*p* < 0.05). In addition, Chao1 and Shannon analysis showed that the diversity and abundance of bacterial communities in the sediment samples were higher than those in the corresponding water samples (*p* < 0.001) (Table 2). Of the 3771 OTUs, 369 were shared by water samples and 1223 were shared by sediment samples from the two lakes (Figure 2B).

According to the results of the OTU classification, 32 phyla were determined in the water samples from the two lakes. Taxonomic analysis revealed that *Proteobacteria* (19.88–45.16%) was the most dominant phylum, followed by *Actinobacteria* (10.48–32.37%), *Bacteroidetes* (2.34–26.33%), and *Chloroflexi* (0.8–22.55%) (Figure 3A). In contrast, *Acidobacteria*, *Verrucomicrobia*, *Armatimonadetes*, *Planctomycetes*, and *Chlorobi* only represented a minor portion (Figure 3A). *Bacteroidetes*, *Verrucomicrobia*, and *Firmicutes* were more abundant in oligotrophic Lake Basomtso than in eutrophic Lake South (*p* < 0.001) (Figure 3B). In contrast, *Actinobacteria* and *Chlorobi* were more abundant in the eutrophic lake compared to the oligotrophic lake (*p* < 0.05) (Figure 3B). In addition, *Acidobacteria*, *Armatimonadetes*, *Gemmatimonadetes*, *Spirochaetes*, and *Nitrospirae* could all be detected in the two lakes with low abundance.

Further analysis indicated that *Burkholderiales*, *Frankiales*, *Acidimicrobiales*, *Sphingobacteriales*, *SubsectionI*, and *Flavobacteriales* all dominated in the two lakes (Figure 3C). The abundance levels of *Sphingobacteriales* in the surface and bottom water samples of oligotrophic Lake Basomtso were 9.28% and 9.14%, respectively, which were higher than that in eutrophic Lake South (3.26%). The surface water in oligotrophic Lake Basomtso contained more *Burkholderiales* (18.21%) than did that in eutrophic Lake South (17.10%). Compared to in eutrophic Lake South, *Frankiales* and *Acidimicrobiales* were more abundant in oligotrophic Lake Basomtso (*p* < 0.01) (Figure 3D).

In the sediments, 53 phyla were determined in the two lakes. The top 15 most abundant phyla contributed up to 90% of the total bacterial community composition (Figure 4). Taxonomic analysis based on the relative abundance showed that *Proteobacteria* (35.00%) was the most predominant phylum, followed by *Chloroflexi* (13.58–22.43%), *Acidobacteria* (8.40–14.26%), *Actinobacteria* (7.07–15.06%), *Nitrospira* (2.31–6.43%), and *Bacteroidetes* (2.28–5.58%) (Figure 4A). In contrast, *Planctomycetes*, *Gemmatimonadetes*, *Verrucomicrobia*, and *Nitrospinae* represented a minor portion of the total bacterial community composition (*p* < 0.05) (Figure 4B). In addition, other dominant phyla in the sediment samples included *Armatimonadetes, Cyanobacteria, Firmicutes, Latescibacteria, Planctomycetes,* etc.

*Proteobacteria* was the most abundant bacterial phylum in the sediment samples. *Xanthomonadales*, *Syntrophobacterales*, *Myxococcales*, *SC-I-84*, *Rhodocyclales*, *Burkholderiales*, and *Rhizobiales* were all detected in the sediment samples derived from the two lakes (Figure 4C). In addition, both *Rhizobiales* and *Xanthomonadales* in the sediments showed significant differences between the eutrophic and oligotrophic lakes (*p* < 0.001) (Figure 4D). *Rhizobiales* showed a higher abundance (9.03%) in the oligotrophic lake compared to the eutrophic lake, while *Xanthomonadales* mainly dominated (13.1%) in the eutrophic lake (Figure 4C). Among the unclassified divisions, the abundances of *Nitrospira* and *KD4-96* were both higher in the eutrophic lake than in the oligotrophic lake, whereas *Anaerolineales*, *Xanthomonadales*, *SC-I-84*, and *Syntrophobacterales* were more abundant in the eutrophic lake than in the oligotrophic lake (Figure 4C). In addition, other orders, such as *Sphingobacteriales*, *Micrococcales*, *Sphingomonadales*, *Syntrophobacterales*, *TRA3-20*, and *SZB30*, could also be detected in the sediment samples derived from the two lakes (Figure 4C).

### 3.4. Comparative Analysis of the Bacterial Communities in the Water and Sediment

In oligotrophic Lake Basomtso, a total of 13 bacterial phyla was detected. As shown in Figure 5, *Proteobacteria* was the most abundant phylum in both water and sediment. *Chloroflexi*, *Acidobacteria*, *Actinobacteria*, and *Bacteroidetes* were also the dominant bacterial groups in both water and sediment (Figure 5A). *Proteobacteria* was more abundant in sediments (34.69–36.55%) than in water (22.37–33.57%). *Chloroflexi* also showed much higher proportions in sediments (13.52–15.16%) than in water (1.08–1.55%) (*p* < 0.001) (Figure 5B). *Acidobacteria* was also more dominant in sediments (12.59–14.32%) than in water (0.9–1.2%) (*p* < 0.001) (Figure 5B). In contrast, a much higher proportion of *Actinobacteria* was found in water (17.04–22.52%) than in sediments (10.48–15.01%). In the oligotrophic lake, certain bacterial phyla such as *Latescibacteria* and *Nitrospinae* were only detected in sediments (Figure 5A).

### 3.5. Principal Co-ordinates Analysis

Principal co-ordinates analysis (PCoA) was recruited to reveal the differences in microbial community patterns between oligotrophic Lake Basomtso and eutrophic Lake South. Two PCoAs explained 81.62% of the total variation in the microbial community structure. The bacterial communities in the water and sediment were clearly separated from each other (Figure 6). In oligotrophic Lake Basomtso, the bacterial community structure in surface water was similar to that in bottom water, but they were separate from that in the water in eutrophic Lake South (Figure 6). In the sediments, the bacterial community structure was similar across the two lakes (Figure 6). 

### 3.6. LEfSe Analysis Based on Community Abundance

Using the relative abundance of OTUs as input data, 1201 bacterial genera could be divided into five distinct groups with a logarithmic linear discriminant analysis (LDA) score (LEfSe uses LDA to estimate the effect size of abundance difference of each species) no less than 4.0. As shown in Figure 7, the bacterial lineages enriched in the sediments were *Proteobacteria*, *Chloroflexi*, *Nitrospinae*, *Firmicutes*, *Nitrospirae*, *Gemmatimonadetes*, and *Acidobacteria*. However, the enriched bacteria lineages in the water were *Actinobacteria*, *Planctomycetes*, *Verrucomicrobia*, and *Bacteroidetes*. 

Among these bacterial communities, the LDA values for *Anaerolineales*, *Xanthomonadales*, *SC_I_84*, *Rhodocyclales*, *Syntrophobacterales*, and *Sva0485* in eutrophic Lake South sediments were all higher than 4.0 (Figure 8). In addition, the LDA values for *Frankiales*, *Corynebacteriales*, *Sphingomonadales*, and *PeM15* in eutrophic Lake South water were also greater than 4.0 (Figure 8). In oligotrophic Lake Basomtso sediments, only the taxa of *Rhizobiales*, *Nitrosomonadales*, *Gaiellales*, and *Gemmatimonadales* showed high LDA values (>4.0), and the LDA values for the *Acidimicrobiales*, *Sphingobacteriales*, *Burkholderiales*, and *Flavobacteriales* taxa in the water were greater than 4.0 (Figure 8). 

## 4. Discussion

In the present study, both the nutrient levels (TP, TN) and temperature in eutrophic Lake South were significantly higher than those in oligotrophic Lake Basomtso. As we know, excessive nutrients and high temperature can induce the growth and reproduction of planktonic algae and significantly increase the biomass of phytoplankton [20,21,42]. This may be the reason why the density of phytoplankton in eutrophic South Lake was higher than that in oligotrophic Lake Basomtso. Furthermore, the growth and decomposition of algae will not only increase the pH of water but also induce the growth of bacteria [7,9]. This may explain why some bacterial taxa were more abundant in eutrophic Lake South than in oligotrophic Lake Basomtso. However, at the same time, the growth and decomposition of harmful algae (such as *Microcystis aeruginosa*) could release algal toxins, which might lead to the death of some bacterial species [37,43]. This may be the reason why the bacterial diversity in the eutrophic lake was lower than that in the oligotrophic lake. 

In the eutrophic lake, the abundance of *Actinomycetes* was significantly higher than that in the oligotrophic lake. Previous studies have reported that the abundance of *Actinomycetes* is positively correlated with the density of phytoplankton [44,45]. In addition, Pearce et al. reported that nutrient enrichment induced an apparent shift from *β-proteobacteria* to *Actinobacteria* in an Antarctic freshwater lake [46]. Recently, metabolic reconstruction indicated that *Actinomycetes* are facultative aerobes with transporters and enzymes for the use of pentoses, polyamines, and dipeptides [47,48]. These results suggest that the *Actinobacteria* could serve as saprophytic microbes that prefer eutrophic conditions. On the other hand, in the sediment samples, our study found that *Chloroflexi* was more abundant in the eutrophic lake than in the oligotrophic lake. *Chloroflexi* is a photoautotrophic microorganism that is usually associated with dechlorination and could participate in the degradation of organic matter [49,50,51]. This may explain the higher relative abundance of the *Chloroflexi* in sediments from eutrophic lakes.

In the oligotrophic lake, *Verrucomicrobia* and *Acidobacteria* were more abundant than in the eutrophic lake. *Verrucomicrobia* is an obligate or facultative anaerobic bacterium [52,53]. Previous studies have shown that *Verrucomicrobia* prefers to grow in relatively better habitats [52,54]. Additionally, recent studies have shown that *Verrucomicrobia* can degrade sulfate polysaccharides and act as polysaccharide degraders in freshwater systems [55,56]. In this study, *Verrucomicrobia* also exhibited a higher abundance in the waters of oligotrophic Lake Basomtso. These results suggest that *Verrucomicrobia* prefers to inhabit oligotrophic lakes. In the sediments, the abundance of *Acidobacteria* in the oligotrophic lake was significantly higher than that in the eutrophic lake. *Acidobacteria* can inhabit a wide variety of terrestrial and aquatic habitats and is particularly abundant in acidic soils, peatlands, and mineral-iron-rich environments [57,58,59]. Many studies have indicated that the abundance of *Acidobacteria* can increase when the pH value is lower than 5.5 [60,61]. In this study, the pH value in the oligotrophic lake was lower than that in the eutrophic lake, possibly resulting in the higher abundance of *Acidobacteria* in the oligotrophic lake compared to the eutrophic lake.

Comparative analysis between the water and sediments showed that the bacterial diversity and abundance in sediments were higher than those in water. In particular, the abundance of *Chloroflexi* in sediments was higher than that in water. Previous studies have also reported the detection of a high abundance of *Chloroflexi* in the sediments of many freshwater lakes [62,63,64]. *Chloroflexi* is a photoautotrophic microorganism, usually linked to dechlorination, and can participate in the degradation of organic matter [49,50,51]. Interestingly, our present study also found that *Chloroflexi* was more abundant in the sediment from the eutrophic lake than in the oligotrophic lake, which further confirmed that *Chloroflexi* prefers saprophytic conditions.

In summary, our results displayed significant differences in the bacterial and phytoplankton communities between the oligotrophic lake and the eutrophic lake. Firstly, the phytoplankton density and diversity in the eutrophic lake were both higher than those in the oligotrophic lake. Secondly, the bacterial abundance in the eutrophic lake was higher than that in the oligotrophic lake, but the bacterial community diversity in the oligotrophic lake was higher than that in the eutrophic lake. Thirdly, the abundance of *Actinomycetes* and *Chloroflexi* in the eutrophic lake was significantly higher than that in the oligotrophic lake; in contrast, both *Verrucomicrobia* and *Acidobacteria* were more abundant in the oligotrophic lake compared to the eutrophic lake. Finally, both the bacterial abundance and diversity in the sediment were higher than those in the water in the two different trophic lakes. These results will further enrich our knowledge on phytoplankton and bacterial community structures in different trophic lakes. 

## Figures and Tables

**Figure 1 microorganisms-07-00621-f001:**
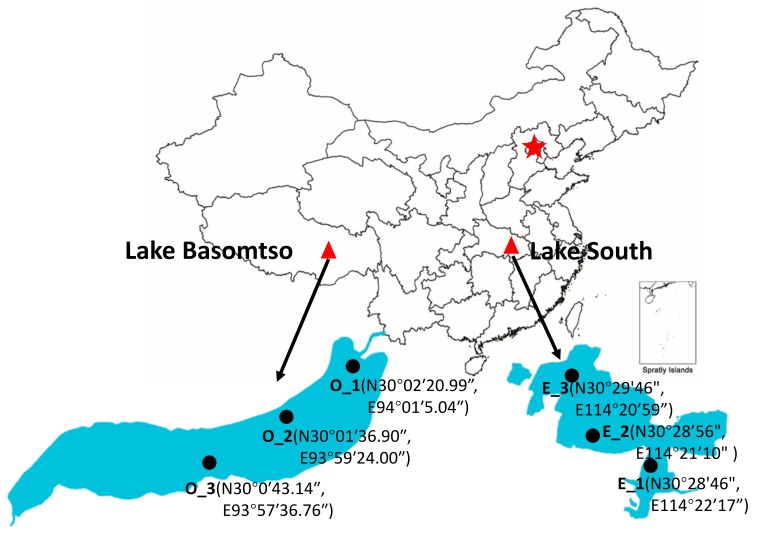
Sampling sites (black circles) and locations of the studied lakes (red triangles) in China. O: oligotrophic Lake Basomtso; E: eutrophic Lake South.

**Figure 2 microorganisms-07-00621-f002:**
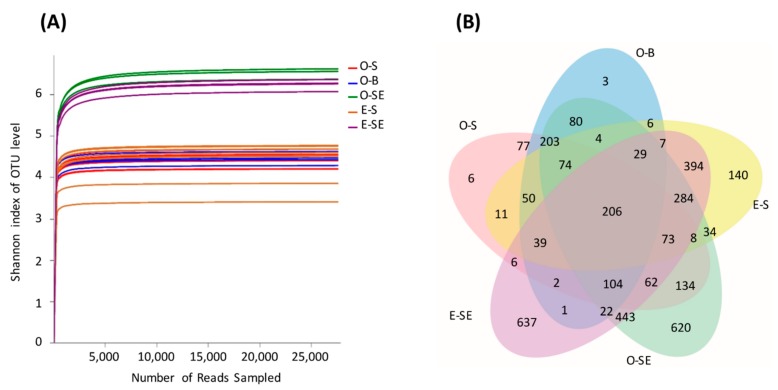
(**A**) Rarefaction curves base on high-throughput sequencing. (**B**) A Venn diagram of shared operational taxonomic units (OTUs). O-S: oligotrophic lake, surface waters; O-B: oligotrophic lake, bottom waters; O-SE: oligotrophic lake, sediments; E-S: eutrophic lake, surface waters; E-SE: eutrophic lake, sediments.

**Figure 3 microorganisms-07-00621-f003:**
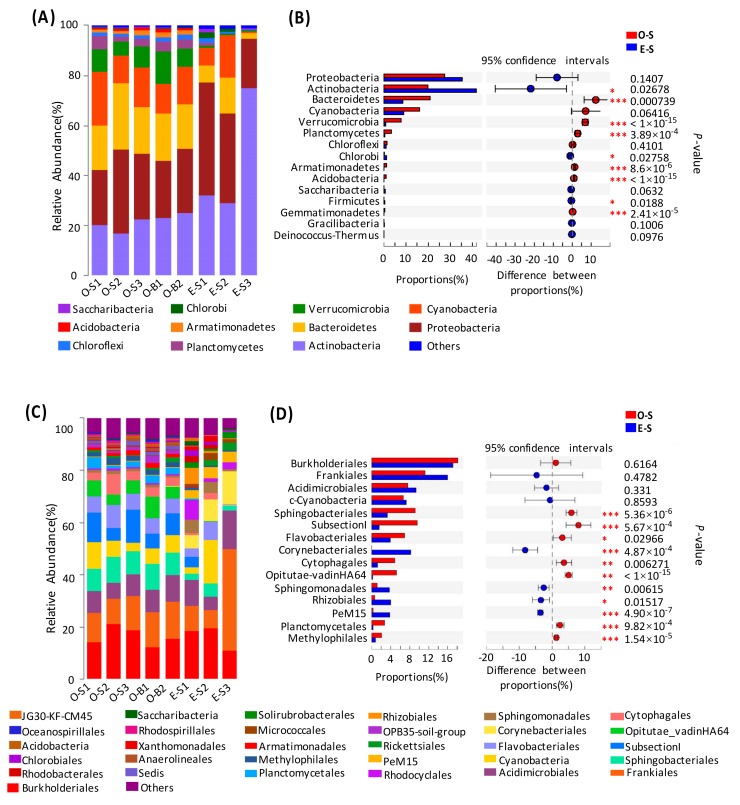
Classification structure and relative abundance in each water sample from Lake Basomtso and Lake South (**A**) at the phylum level and (**C**) at the order level. At the specific level, “Others” means those that account for less than 1% of the total OTUs in each sample. In the overall distribution of bacteria at the phylum level (**B**) and at the order level (**D**) in each water sample, the circle values represent the differences between the proportions in Lake Basomtso (red color) and Lake South (blue color). The bar graph on the left represents the proportion of each bacterial phylum’s abundance in the samples. The difference in bacterial abundance was significant with a *p*-value of <0.05. * *p* < 0.05; ** *p* < 0.01; *** *p* < 0.001. O-S: oligotrophic lake, surface waters, O-B: oligotrophic lake, bottom waters, E-S: eutrophic lake, surface waters.

**Figure 4 microorganisms-07-00621-f004:**
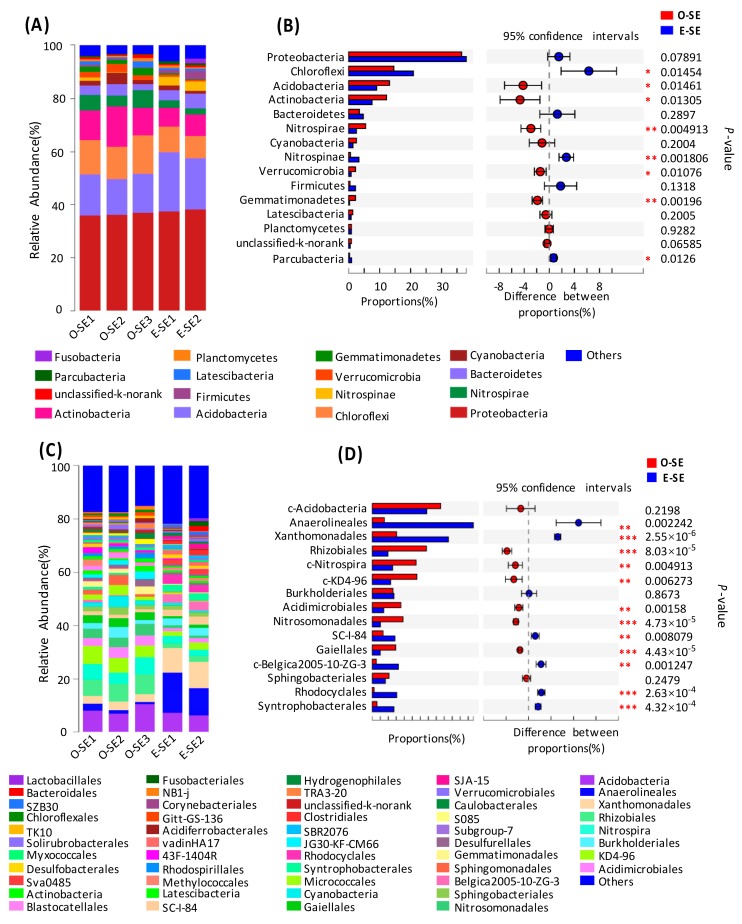
Classification structure and relative abundance in each sediment sample from Lake Basomtso and Lake South (**A**) at the phylum level and (**C**) at the order level. At the specific level, “Others” means those that account for less than 1% of the total OTUs in each sample. In the overall distribution of bacteria at the phylum level (**B**) and at the order level (**D**) in each sediment sample, the circle values represent the differences between the proportions in Lake Basomtso (red color) and Lake South (blue color). The bar graph on the left represents the proportion of each bacterial phylum’s abundance in the samples. The difference in bacterial abundance was significant with a *p*-value of <0.05. * *p* < 0.05; ** *p* < 0.01; *** *p* < 0.001. O-SE: oligotrophic lake, sediments, E-SE: eutrophic lake, sediments.

**Figure 5 microorganisms-07-00621-f005:**
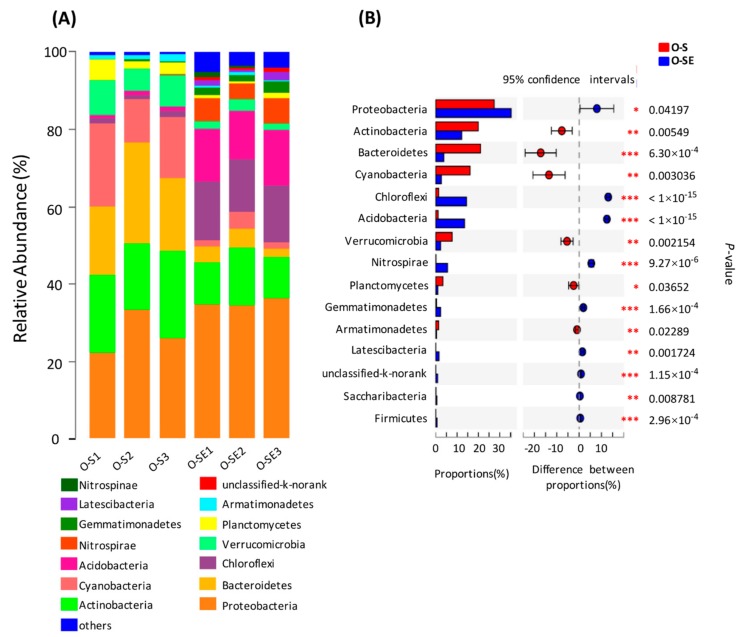
Distribution of major bacterial groups (**A**) across the sediment and water samples and (**B**) overall distribution between the sediment and water samples in oligotrophic Lake Basomtso. The circle values represent the differences between the proportions in water samples (red color) and sediment samples (blue color) from Lake Basomtso. Bars on the left represent the proportion of each bacterial phylum’s abundance in the samples. Bacterial abundance differences with a *p*-value of <0.05 were considered significant. * *p* < 0.05; ** *p* < 0.01; *** *p* < 0.001. O-S: oligotrophic lake, surface waters, O-SE: oligotrophic lake, sediments.

**Figure 6 microorganisms-07-00621-f006:**
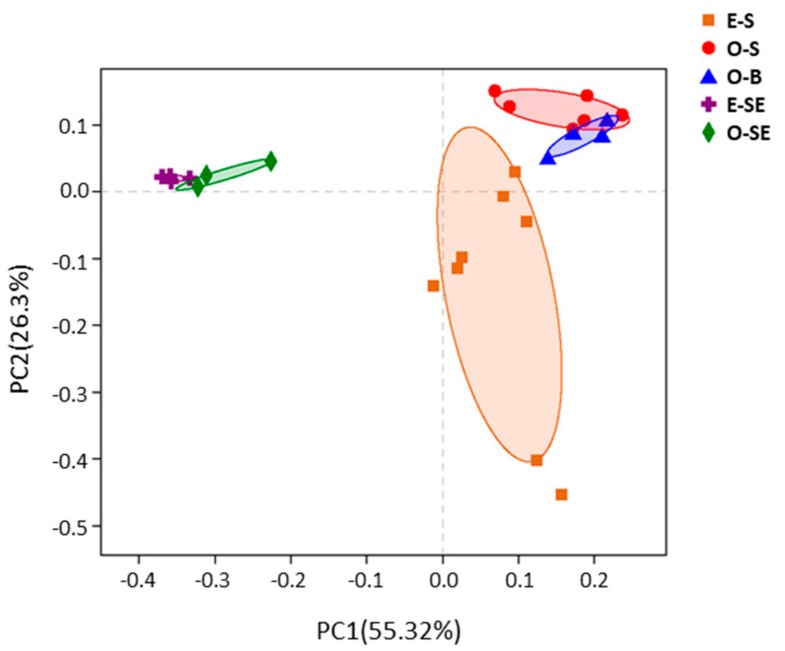
Ordination plot showing the grouping of the water and sediment samples from the two lakes according to their bacterial community structure; the principle coordination was analyzed based on the Bray–Curtis distance matrix. O-S: oligotrophic lake, surface waters, O-B: oligotrophic lake, bottom waters, O-SE: oligotrophic lake, sediments, E-S: eutrophic lake, surface waters, E-SE: eutrophic lake, sediments.

**Figure 7 microorganisms-07-00621-f007:**
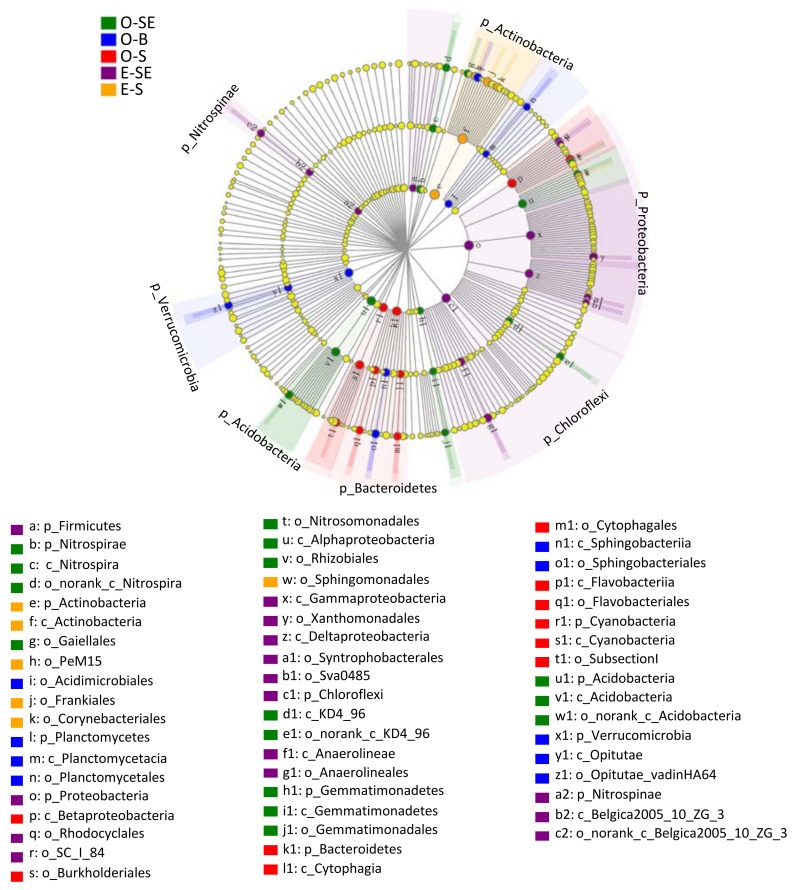
Cladogram showing the phylogenetic distribution of microbial communities associated with the water and sediment samples from the two lakes; lineages with linear discriminant analysis (LDA) values of 4.0 or higher were determined by LEfSe. Differences are represented by the color of the most abundant class. Green (a) represents Lake Basomtso sediments, blue (b) represents Lake Basomtso bottom water samples, red (c) represents Lake Basomtso surface water samples, purple (d) represents Lake South sediments, orange (e) represents Lake South water samples, and yellow (f) represents an insignificant difference. The diameter of each circle is proportional to a taxon’s abundance. Circles from the inner region to outer region represent the phylogenetic levels from domain to genus. O-S: oligotrophic lake, surface waters, O-B: oligotrophic lake, bottom waters, O-SE: oligotrophic lake, sediments, E-S: eutrophic lake, surface waters, E-SE: eutrophic lake, sediments.

**Figure 8 microorganisms-07-00621-f008:**
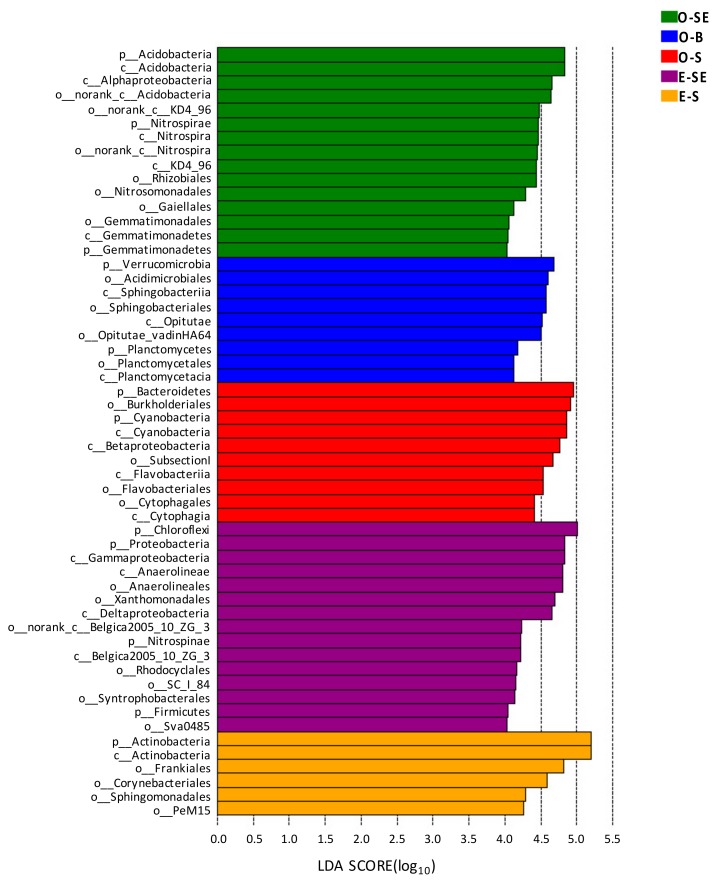
Microbial indicator groups in the two lakes with LDA >4.0. O-S: oligotrophic lake, surface waters, O-B: oligotrophic lake, bottom waters, O-SE: oligotrophic lake, sediments, E-S: eutrophic lake, surface waters, E-SE: eutrophic lake, sediments.

**Table 1 microorganisms-07-00621-t001:** Environmental parameters and phytoplankton density in oligotrophic Lake Basomtso and eutrophic Lake South.

Environmental Parameters	Oligotrophic (Lake Basomtso)	Eutrophic (Lake South)
T (°C)	12.6 ± 0.62	29.4 ± 0.08 ***
pH	7.42 ± 0.07	8.80 ± 0.06 ***
DO (mg/L)	7.46 ± 0.22	8.45 ± 0.60
TN (mg/L)	1.62 ± 0.97	6.73 ± 0.24 **
TP (mg/L)	0.01 ± 0.002	0.13 ± 0.04 **
Chl-*a* (μg/L)	0.60 ± 0.08	220.54 ± 18.20 ***
Total phytoplankton density (cells/L)	3933	4.82 × 10^8^

Noted: T: temperature; DO: dissolved oxygen; TN: total nitrogen; TP: total phosphorus; Chl-*a*: chlorophyll *a.* ** *p* < 0.01; *** *p* < 0.001.

**Table 2 microorganisms-07-00621-t002:** Bacterial abundance and diversity in eutrophic Lake South and oligotrophic Lake Basomtso.

Sample Nature	Sample ID	Assigned Reads	97% Similarity			
			OTUs	Shannon	Chao1	Coverage
Water	O-S1	4,1086	518	3.426241	239.5462	0.998772
	O-S2	3,6344	587	3.753438	275.2587	0.998930
	O-S3	4,3431	514	3.470968	244.3997	0.998709
	O-B1	4,1275	556	3.652926	258.7321	0.998741
	O-B2	3,7534	519	3.532502	230.0804	0.998993
	E-S1	3,1767	636	3.837348	281.7246	0.998898
	E-S2	3,1891	632	3.548155	318.5813	0.998657
	E-S3	4,4184	441	2.575510	209.3030	0.998883
Sediment	O-SE1	3,3418	1885	6.539644	2041.603	0.992609
	O-SE2	2,7443	1659	6.419282	1806.995	0.990854
	O-SE3	3,3654	1545	6.337208	1675.043	0.994295
	E-SE1	3,0407	1941	6.274548	1914.850	0.992344
	E-SE2	3,6831	1889	6.129345	1902.770	0.994143

Notes: O-S1, O-S2, and O-S3: Oligotrophic Lake Basomtso, surface water samples; O-B1 and O-B2: Oligotrophic Lake Basomtso, bottom water samples; O-SE1, O-SE2, and O-SE3: Oligotrophic Lake Basomtso, sediment samples; E-S1, E-S2, and E-S3: Eutrophic Lake South, water samples; E-SE1 and E-SE2: Eutrophic Lake South, sediment samples.

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
