# Peer review of "Phytoplankton and Bacterial Community Structure in Two Chinese Lakes of Different Trophic Status"

_microorganisms, 2019, doi:10.3390/microorganisms7120621_

Round 1
Reviewer 1 Report
This manuscript presents field study on bacterial, phytoplankton and zooplankton taxonomic composition in two lakes of low and high trophic conditions. The topic is interesting, but the manuscript has some shortcomings; it is not well organized and discussed. Many sentences are unclear and require reformulation. Specific examples are below.
Some specific comments:
Title
The title should be slightly corrected; I would suggest something like this: “Phytoplankton and bacterial community structure in two Chinese lakes of different trophic status", or “Comparative study of phytoplankton and bacterial community between oligotrophic lake and eutrophic lake in China”
Abstract
It is known that each group of aquatic organisms (phytoplankton, bacteria, autotrophic picoplankton, nanoflagellates, ciliates, rotifers, crustaceans, fish) plays an important role in aquatic ecosystems; please specify the role of bacteria and phytoplankton in freshwater ecosystems.
The names of the lakes say little to the readers, it is better to use the trophy of lakes, all the more so because the authors describe only two lakes with contrasting trophy (oligotrophic and eutrophic).
Line 16: change “played” to “play”.
Line 17: should be “bacterial and phytoplankton communities”
Line 18: change “Basomtso Lake and South Lake” to “Lake Basomtso and Lake South” - please correct this throughout the manuscript. Remove “respectively”.
Line 20: Cyanobacteria are the component of phytoplankton
Line 24: “were significantly abundant..” - not sure what you are saying here.
Line 26: change “were only detected in” to “were identified only in”
Line 28: change “oligotrophic lakes and eutrophic lakes” to “oligotrophic lake and eutrophic lake”
Line 31: The authors write that “...there was a negative correlation between”, but in the M&M and Results sections it is not mentioned how such analyses were performed. In addition, dataset (three data for surface waters and sediments, and two data for bottom waters for each lake) does not give the possibility to perform correlations.
Introduction
The Introduction should be developed further; an overview of the literature on the taxonomic composition of bacteria and phytoplankton in lakes of different trophic status should be added here to emphasize the aim of the study. There are no clear hypotheses tested.
The description of lakes (lines 58-65) should be in M&M section, in subsection “Study sites”
Line 37: remove “respectively”
Line 41: remove “flora”
Line 42: remove “as the oldest biota”
Line 44: change “As an important part of the microbial community, bacteria play a vital..” to “Bacteria, as an important part of the microbial community, play a vital…”.
Line 45: change “ecosystem” to “ecosystems”
Line 47: remove “the different”
Line 48: change “states” to “status”
Line 52: what is meant by the phrase “roudly grasp”?
Lines 57-58: remove this sentence
Line 67: remove “respectively”
Materials and Methods
Morphometric characteristics (area, mean and maximum depth) of the lakes should be added, e.g. in table.
Your acronyms (BL-SW1 and so on) could be shorter; I would suggest something like this: O-S1, O-S2, O-S3 (oligotrophic Basomtso Lake, the surface water samples), O-B1, O-B2 (the bottom water samples), O-SE1, O-SE2, O-SE3 (the sediment samples),etc.
This section needs more details on phytoplankton, zooplankton, diversity indices and statistical analyses.
Line 74: change “experiment: to “study”
Line 82: change “into a 5 L” to “with a”
Line 83: change “detected by” to “measured”
Line 85: change “deposited” to “stored”
Line 86: change “in 4°C incubator” to “at 4°C in incubator”
Line 87: change “filtered by a Millipore filter membrane with an aperture of 5 μm” to “filtered through 5.0 µm pore size membrane filters (Millipore)”
Line 88: should be “collected on 0.22 µm pore size membrane filters (Millipore)” and “The filters were”
Line 90: change “processing” to “procedures”
Line 114: “Raw fastq files” – not sure what you are saying here.
Results
The results, especially those regarding bacterial composition, are extensive and difficult to follow.I do not think is necessary to divide in so many subsections.
Temperature, oxygen, TN, TP, Chl-a are given in Table 1 – please avoid to repeat the results.
The authors write that “there were significant differences in the physic-chemical characteristics between South Lake and Basomtso Lake”, however there are no information which test was used to determine differences between lakes.
The authors found that, for example, “Bacillariophyta were the dominant species (70.4%), followed by Cyanobacteria (14.8%), Chlorophyta (3.7%), Euglenophyta (3.7%), Pyrrophyta (3.7%) and Chrysophyta (3.7%).“ – what is the percentage of? numbers, biomass?
Many phrases are unclear, for example: “Taxonomic assignment”, “conducted based on”, “a large bacterial community”, “the South Lake has the highest species richness index value”, “lowly detected”, “each differentially abundant feature”.
Line 146: “Chlorophyta was the dominant species (75.9%)” – chlorophyta are a phylum, not species, similarly “Diaptomidae were the dominant species” – diaptomidae are a family, not species (see your Tables 1 and 3).
Lines 156-158: remove this sentence.
Lines 163-165 should be in M&M section.
Line 218: change “basing” to “based”
Line 229-230: change “in the sediment samples were showed significantly differences” to ”
“in the sediment showed significant differences”
Line 308: remove “were”
Discussion
The Discussion should be developed further. There is no discussion about bacteria in relation to other studied physical, chemical and biological factors. The studied lakes differed clearly in water temperature that can strongly affect the abundance and structure of phytoplankton and bacterial communities.
Remove the first two sentences – they are in the “Introduction” section.
Line 326: change “flora” to “group”
Line 329: “abundantly distributed” - re-word phrase.
Line 331: “oligotrophic groups” – unclear, maybe you mean “the α-proteobacteria are characteristic of oligotrophic lakes”
Line 335: change “Actinobacteria is” to „ Actinobacteria are”
Line 338: “eutrophication lakes” – may be you mean “eutrophic lakes”
Lines 340, 349, 353, 356: “more abundantly detected”, “poorly detected”, “highly detected” – incorrect phrases.
Lines 345-348: re-word this sentence and divide into several ones.
Lines 360-361: remove this sentence.
Lines 375-377 (Author Contributions) – I did not find the initials LXJ and LDP; I suppose that HM and DC should be replaced by MH and CD.
References
Lines 437-439: “Pseudoalteromonas, Proteobacteria, Chattonella, Gymnodinium, Heterosigma” should be in italics.
Figure 1. caption: change “sampling” to “studied”
Figs 3, 4, 5 captions: change “Other” to “Others”, What is meant by the phrases “the South Lake samples have a positive difference”, “a negative difference”? I have never seen it before.
Tables 1 and 2 should be combined into one table to better visualize the differences in the species composition between the studied lakes (see below). The same applies to Tables 3 and 4. Moreover, I think that Family, Order, Genus are unnecessary; in the case of zooplankton, give the species name, e.g. Brachionus sp.
| Species |
Oligotrophic Lake Basomtso |
Eutrophic Lake South |
| Bacillariophyta | ||
| Cyclotella meneghiniana | X | X |
| Cyclotella aslerocastata | X | |
| Total species number | 27 | 86 |
Table 1: Oscillatoriales are not a genus, but order
Table 2: “Asterionella”, “Trachelomonas” instead of “Asterionelle”, “Tiachelomonas”
Table 3: Diaptomidae are not a genus; if some individuals have not been identified, give sp.1, sp. 2, ...
Author Response
Reviewer 1
Comments and Suggestions for Authors
This manuscript presents field study on bacterial, phytoplankton and zooplankton taxonomic composition in two lakes of low and high trophic conditions. The topic is interesting, but the manuscript has some shortcomings; it is not well organized and discussed. Many sentences are unclear and require reformulation. Specific examples are below.
Response:
Thank you very much for your help with the paper review. As you suggested, we have revised our manuscript as follows.
Some specific comments:
Title
The title should be slightly corrected; I would suggest something like this: “Phytoplankton and bacterial community structure in two Chinese lakes of different trophic status", or “Comparative study of phytoplankton and bacterial community between oligotrophic lake and eutrophic lake in China”
Response:
Thank you very much for your suggestions. As you suggested, we have changed the title to “Phytoplankton and bacterial community structure in two Chinese lakes of different trophic status”.(Line 2-3)
Abstract
It is known that each group of aquatic organisms (phytoplankton, bacteria, autotrophic picoplankton, nanoflagellates, ciliates, rotifers, crustaceans, fish) plays an important role in aquatic ecosystems; please specify the role of bacteria and phytoplankton in freshwater ecosystems.
Response:
Thank you very much for your suggestions. As you suggested, we have revised the abstract (Line 15-30).
“Phytoplankton are the primary producers at the basis of aquatic food webs, and bacteria play an important role in energy flow and biochemical cycling in aquatic ecosystems.”(Line 15-16)
The names of the lakes say little to the readers, it is better to use the trophy of lakes, all the more so because the authors describe only two lakes with contrasting trophy (oligotrophic and eutrophic).
Response:
Thank you very much for your suggestions. As you suggested, we have used the oligotrophic and eutrophic lake to respectively represent the Lake Basomtso and Lake South in the revised abstract and manuscript.
Line 16: change “played” to “play”.
Response:
Thank you very much for your suggestions. As you suggested, we have corrected “played” to “play”. (Line 16)
Line 17: should be “bacterial and phytoplankton communities”
Response:
Thank you very much for your suggestions. As you suggested, we have corrected “bacteria and phytoplankton communities” to “bacterial and phytoplankton communities”. (Line 17)
Line 18: change “Basomtso Lake and South Lake” to “Lake Basomtso and Lake South” - please correct this throughout the manuscript. Remove “respectively”.
Response:
Thank you very much for your suggestions. As you suggested, we have corrected “Basomtso Lake and South Lake” to “Lake Basomtso and Lake South” throughout this manuscript (Line 18, 19, 25….), and removed “respectively” in Line18.
Line 20: Cyanobacteria are the component of phytoplankton
Response:
Thank you very much for your suggestions. we have removed “Cyanobacteria”.
Line 24: “were significantly abundant.” - not sure what you are saying here.
Response:
Thank you very much for your suggestions. It’s my mistake. We have removed “significantly” in Line 25.
Line 26: change “were only detected in” to “were identified only in”
Response:
Thank you very much for your suggestions. As you suggested, we have corrected “were only detected in” to “were identified only in” in Line 28.
Line 28: change “oligotrophic lakes and eutrophic lakes” to “oligotrophic lake and eutrophic lake”
Response:
Thank you very much for your suggestions. As you suggested, we have corrected “oligotrophic lakes and eutrophic lakes” to “oligotrophic lake and eutrophic lake”. (Line 25-26)
Line 31: The authors write that “...there was a negative correlation between”, but in the M&M and Results sections it is not mentioned how such analyses were performed. In addition, datas et (three data for surface waters and sediments, and two data for bottom waters for each lake) does not give the possibility to perform correlations.
Response:
Thank you very much for your comments. You are right. The conclusion “there was a negative correlation between the phytoplankton diversity and bacterial diversity in freshwater ecosystems” is not suitable in this study. As you suggested, we have removed this conclusion and revised our abstract. (Line 15-30)
“Abstract: Phytoplankton are the primary producers at the basis of aquatic food webs, and bacteria play an important role in energy flow and biochemical cycling in aquatic ecosystems. In this study, both the bacterial and phytoplankton communities were examined in the oligotrophic Lake Basomtso (China) and the eutrophic Lake South (China). Microscopic examination showed that the phytoplankton density and diversity in eutrophic lake were higher than those in oligotrophic lake. Furthermore, Chlorophyta (68.26%) and Cryptophyta (24.48%) were the dominant groups in eutrophic lake, while Bacillariophyta (94.96%) were dominated in oligotrophic lake. The bacterial communities in the waters and sediments of the two lakes were mainly composed of Proteobacteria (mean of 32.05%), Actinobacteria (mean of 24.64%), Bacteroidetes (mean of 11.93%), and Chloroflexi (mean of 6.12%). Comparative analysis showed that the abundance of bacteria in eutrophic lake were higher than that in oligotrophic lake (P<0.05), but the bacterial diversity in oligotrophic lake was higher than that in eutrophic lake (P<0.05). Finally, the bacterial abundance and diversity in the sediments of the two lakes were higher than those in the water samples (P<0.05), and the Latescibacteria and Nitrospinae groups were identified only in the sediments. These results, taken together, suggested that both phytoplankton and bacterial community were different in the oligotrophic lake and eutrophic lake.”
Introduction
The Introduction should be developed further; an overview of the literature on the taxonomic composition of bacteria and phytoplankton in lakes of different trophic status should be added here to emphasize the aim of the study. There are no clear hypotheses tested.
Response:
Thank you very much for your comments. As you suggested, we have added some information to emphasize the aim of the study. (Line 45-52)
“Phytoplankton are the primary producers at the basis of aquatic food webs, which can promptly respond to environmental changes [11]. In aquatic ecosystems, interactions between phytoplankton and bacteria have been proposed to influence bacterial community [12-14]. Phytoplankton could provide shelter for predation-related bacteria, as well as the habitat for endophytic bacteria living within the algae cells and epiphytic bacteria on the surface of algae [15,16]. In addition to providing the shelter, phytoplankton could provide organic matter (e.g. sugars, amino acids) to bacteria [12]. Interestingly, phytoplankton could also display a negative affect in bacterial community through nutrient competition and antibiotic release [17].”
The description of lakes (lines 58-65) should be in M&M section, in subsection “Study sites”
Response:
Thank you very much for your suggestions. As you suggested, we have transferred this sentence to the M&M section. (Line 70-75)
“2.1 Study sites and sample collection
Lake Basomtso is one of the largest freshwater dammed lake in the east of Tibet, and is a oligotrophic freshwater lake. Its length, area and maximum dept are 18 km, 27 km2 and 120 m, respectively (Fig. 1). Lake South is a typical eutrophic lake in Wuhan City, which is located in the Middle-Lower Yangtze plains. Its area and average depth are 5.50 km2 and 1.6 m, respectively (Fig. 1).”
Line 37: remove “respectively”
Response:
Thank you very much for your suggestions. As you suggested, we have removed “respectively” in Line 35.
Line 41: remove “flora”
Response:
Thank you very much for your suggestions. As you suggested, we have removed “flora” in Line 38-39.
Line 42: remove “as the oldest biota”
Response:
Thank you very much for your suggestions. As you suggested, we have removed “as the oldest biota” and modified the sentence. (Line 38-39).
Line 44: change “As an important part of the microbial community, bacteria play a vital…” to “Bacteria, as an important part of the microbial community, play a vital…”.
Response:
Thank you very much for your suggestions. As you suggested, we have corrected “As an important part of the microbial community, bacteria play a vital…” to “Bacteria, as an important part of the microbial community, play a vital…” in Line 42-45.
Line 45: change “ecosystem” to “ecosystems”
Response:
Thank you very much for your suggestions. As you suggested, we have corrected “ecosystem” to “ecosystems” in Line 41.
Line 47: remove “the different”
Response:
Thank you very much for your suggestions. As you suggested, we have removed “the different”.
Line 48: change “states” to “status”
Response:
Thank you very much for your suggestions. This sentence has been removed.
Line 52: what is meant by the phrase “roundly grasp”?
Response:
Thank you very much for your comments. Sorry for our shortcoming, we have modified this sentence as “In recent years, this technology has been widely used in early detection of aquatic invasion and investigation of water biodiversity” in Line 55-57.
Lines 57-58: remove this sentence
Response:
Thank you very much for your suggestions. As you suggested, we have removed “China has many natural lakes, most of them scattered in the Qinghai-Tibet Plateau and the Middle-Lower Yangtze plains”.
Line 67: remove “respectively”
Response:
Thank you very much for your suggestions. As you suggested, we have removed “respectively” in Line 65.
Materials and Methods
Morphometric characteristics (area, mean and maximum depth) of the lakes should be added, e.g. in table.
Response:
Thank you very much for your suggestions. As you suggested, we have added study sites. (Line 71-75)
“Lake Basomtso is one of the largest freshwater dammed lake in the east of Tibet, and is a oligotrophic freshwater lake. Its length, area and maximum dept are 18 km, 27 km2 and 120 m, respectively (Fig. 1). Lake South is a typical eutrophic lake in Wuhan City, which is located in the Middle-Lower Yangtze plains. Its area and average depth are 5.50 km2 and 1.6 m, respectively (Fig. 1).”
Your acronyms (BL-SW1 and so on) could be shorter; I would suggest something like this: O-S1, O-S2, O-S3 (oligotrophic Basomtso Lake, the surface water samples), O-B1, O-B2 (the bottom water samples), O-SE1, O-SE2, O-SE3 (the sediment samples),etc.
Response:
Thank you very much for your suggestions. As you suggested, we have revised this sentence. (Line 76-80)
“O-S1, O-S2, O-S3 (oligotrophic Basomtso Lake, the surface water samples), O-B1, O-B2 (the bottom water samples), and O-SE1, O-SE2, O-SE3 (the sediment samples) were all collected from river inlet to lake center (Fig. 1); similarly, E-S1, E-S2, E-S3 (eutrophic South Lake, the water samples), E-SE1, E-SE2 (the sediment samples) were collected from the shore to the center (Fig. 1)”
This section needs more details on phytoplankton, zooplankton, diversity indices and statistical analyses.
Response:
Thank you very much for your comments. Sorry for our shortcoming. As you suggested, we have added the methods for plankton collection and analysis in the revised manuscript. (Line 95-106). In addition, the statistical analyses have also been added in the Materials and Methods. (Line 140-144)
Details on phytoplankton and zooplankton
“2.2 Qualitative and quantitative analyses of plankton
The phytoplankton samples were collected by using 1L sampler, and the samples were fixed in Lugol solution. Then, the fixed samples were transported to the laboratory and precipitated by DXCJQ-1L plankton settler within 48 h. Finally, 50 mL concentrated sample was secondary fixed by using 1mL formaldehyde. The zooplankton samples were collected by using the 5 L sampler. Then, the samples were filtered through the 13# plankton net and fixed in formaldehyde solution. Using a 10 mL sedimentation chamber, the qualitative and quantitative analysis of plankton in oligotrophic lakes was performed by inverted microscope technique with 100x and 400x magnifications, respectively. The phytoplankton and zooplankton samples in the eutrophic lake were qualitatively and quantitatively counted by an Optical Microscope using 0.1 mL and 1 mL counting chamber, respectively. Besides, species identification was performed as previous studies [24-26]”. (Line95-106)
Statistical analyses
“The three replicates were expressed as Mean ± SD; all data were tested for normality of distribution using the Shapiro-Wilk normality test. Then, one-way ANOVA was used to test the significant difference according to different experiment groups. The differences between groups were considered as significant at P<0.05 (“*”), highly significant at P<0.01 (“**”) or at P<0.001 (“***”)”. (Line 140-144)
Line 74: change “experiment: to “study”
Response:
Thank you very much for your suggestions. As you suggested, we have corrected “experiment” to “study”. (Line 75)
Line 82: change “into a 5 L” to “with a”
Response:
Thank you very much for your suggestions. As you suggested, we have corrected “into a 5 L” to “with a”. (Line 82)
Line 83: change “detected by” to “measured”
Response:
Thank you very much for your suggestions. As you suggested, we have corrected “detected by” to “measured”. (Line 84)
Line 85: change “deposited” to “stored”
Response:
Thank you very much for your suggestions. As you suggested, we have corrected “deposited” to “stored”. (Line 86)
Line 86: change “in 4°C incubator” to “at 4°C in incubator”
Response:
Thank you very much for your suggestions. As you suggested, we have corrected “in 4°C incubator” to “at 4°C in incubator”. (Line 86)
Line 87: change “filtered by a Millipore filter membrane with an aperture of 5 μm” to “filtered through 5.0 µm pore size membrane filters (Millipore)”
Response:
Thank you very much for your suggestions. As you suggested, we have corrected “filtered by a Millipore filter membrane with an aperture of 5 μm” to “filtered through 5.0 µm pore size membrane filters (Millipore)”. (Line 88)
Line 88: should be “collected on 0.22 µm pore size membrane filters (Millipore)” and “The filters were”
Response:
Thank you very much for your suggestions. As you suggested, we have corrected “collected by a Millipore filter membrane with an aperture of 0.22 μm” to “collected on 0.22 μm pore size membrane filters (Millipore)” in Line 90, and corrected “The membrane was” to “The filters were” in Line 91
Line 90: change “processing” to “procedures”
Response:
Thank you very much for your suggestions. As you suggested, we have corrected “processing” to “procedures” in Line 92.
Line 114: “Raw fastq files” – not sure what you are saying here.
Response:
Thanks a lot for your comments. Sorry for our mistake, we have corrected “Raw fastq files” to “Raw reads” in Line 127.
Results
The results, especially those regarding bacterial composition, are extensive and difficult to follow. I do not think is necessary to divide in so many subsections.
Response:
Thank you very much for your suggestions. I am sorry for the confusing in our results. We have corrected the title in each subsection. In addition, as you suggested, the Result 3.3, 3.4 and 3.5 has been merged together, named as “3.3 Bacterial community structure in the two lakes”. (Line 179)
3.1. General statistical information of environmental variables
3.2. Plankton community composition in the two lakes
3.3. Bacterial community strucuture in the two lakes
3.4. Comparative analysis of bacterial community in the water and sediment
3.5. Principal co-ordinates analysis
3.6. LEfSe analysis based on community abundance
Temperature, oxygen, TN, TP, Chl-a are given in Table 1 – please avoid to repeat the results.
Response:
Thank you very much for your suggestions. As you suggested, we have deleted the repetition in Line 147-157.
The authors write that “there were significant differences in the physic-chemical characteristics between South Lake and Basomtso Lake”, however there are no information which test was used to determine differences between lakes.
Response:
Thank you very much for your comments. As you suggested, in the revised Table 1, the three replicates were expressed as Mean ± SD. Then, one-way ANOVA was used to test the significant difference according to different experiment groups. The differences between groups were considered as highly significant at P<0.01 (“**”) or at P<0.001 (“***”). (Table 1)
In addition, the p values were also added in the result 3.1 subsection. (Line 151 & 152 &154 & 155)
The authors found that, for example, “Bacillariophyta were the dominant species (70.4%), followed by Cyanobacteria (14.8%), Chlorophyta (3.7%), Euglenophyta (3.7%), Pyrrophyta (3.7%) and Chrysophyta (3.7%). “– what is the percentage of? numbers, biomass?
Response:
Thank you very much for your suggestions. Sorry for our shortcoming. In the original manuscript, it is the percentage of numbers. In the revised manuscript, we have represented the data with the percentage of phytoplankton density. (Line 163-167; Table1)
“Microscopic examination showed that the total phytoplankton density in eutrophic Lake South (4.82×108 cells/L) was higher than that in oligotrophic Lake Basomtso (3933 cells /L) (Table 1). Among them, Bacillariophyta accounted for 94.96% of total phytoplankton density in oligotrophic lake, whereas the Chlorophyta and Cyanophyta accounted for 68.26% and 24.48% of total phytoplankton density in eutrophic Lake South, respectively.”
Many phrases are unclear, for example: “Taxonomic assignment”, “conducted based on”, “a large bacterial community”, “the South Lake has the highest species richness index value”, “lowly detected”, “each differentially abundant feature”.
Response:
Thank you very much for your comments. Sorry for our shortcoming. As you suggested, these unclear phrases have been corrected. (Line 180-189)
Line 146: “Chlorophyta was the dominant species (75.9%)” – chlorophyta are a phylum, not species, similarly “Diaptomidae were the dominant species” – diaptomidae are a family, not species (see your Tables 1 and 3).
Response:
Thank you very much for your comments. Sorry for our shortcoming. As you suggested, these errors have been corrected in the revised manuscript. (Line 165-178)
“Among them, Bacillariophyta accounted for 94.96% of total phytoplankton density in oligotrophic lake, whereas the Chlorophyta and Cyanophyta accounted for 68.26% and 24.48% of total phytoplankton density in eutrophic Lake South, respectively. In addition, there were 22 genera, 16 families and 6 phylum phytoplankton in Lake Basomtso, while the phytoplankton in Lake South was composed of 6 phylums, 23 families and 36 genera (Table S1). Among them, the main dominant species were Cyclotella meneghiniana, Nitzschia linearis and Surirella biseriate in oligotrophic Lake Basomtso (Table S1). In contrast, the main dominant species in Lake South were Anabaena circinalis, Anabaena viguieri, Microcystis wesenbergii, and Merismopedia minima (Table S1).
Several zooplankton were detected in Lake Basomtso, including rotifers (4 families and 6 genera) and copepods (3 families and 3 genera) (Table S2). Among them, Diaptomidae were dominanted in Bsomtso Lake (Table S2). In addition, compared to Lake Basomtso, the zooplankton in Lake South were more abundant, including rotifers (5 families and 6 genera), copepods (1 family and 1 genera), and cladocerans (1 family and 1 genera) (Table S2). Among them, the rotifers were the most abundant community in Lake South (Table S2). ”
Lines 156-158: remove this sentence.
Response:
Thank you very much for your suggestions. As you suggested, we have removed “High-throughput sequencing technique was used to perform the bacterial community diversity in the water and sediment samples from two different trophic lakes, Basomtso Lake (Tibet, China) and South Lake (Wuhan, China), respectively”.
Lines 163-165 should be in M&M section.
Thank you very much for your suggestions. As you suggested, we have transferred the sentence “Non-parametric indicators (Chao1 and Shannon) were used to evaluate the relationship of bacterial community diversity characteristics between Basomtso Lake and South Lake.” to the data analysis of the M&M section. (Line 136)
Line 218: change “basing” to “based”
Response:
Thank you very much for your suggestions. As you suggested, we have corrected “basing” to “based”. (Line 233)
Line 229-230: change “in the sediment samples were showed significantly differences” to “in the sediment showed significant differences”
Response:
Thank you very much for your suggestions. As you suggested, we have corrected “in the sediment samples were showed significantly differences” to “in the sediment showed significant differences”. (Line 244)
Line 308: remove “were”
Response:
Thank you very much for your suggestions. As you suggested, we have removed “were”.
Discussion
The Discussion should be developed further. There is no discussion about bacteria in relation to other studied physical, chemical and biological factors. The studied lakes differed clearly in water temperature that can strongly affect the abundance and structure of phytoplankton and bacterial communities.
Response:
Thank you very much for your comments. As you suggested, we have added the discussion about bacteria in relation to other studied physical, chemical and biological factors. (Line 324-334)
“In the present study, both the nutrient (TP, TN) and temperature in eutrophic Lake South were higher than those in oligotrophic Lake Basomtso. As we know, the high excessive nutrient and temperature could induce the growth and reproduction of planktonic algae and significantly increase the biomass of phytoplankton [27]. These maybe the reason why the phytoplankton density in eutrophic South Laker were higher than that in oligotrophic Lake Basomtso. Furthermore, the growth and decomposition of algae will not only increase the pH of water, but also induce the growth of bacteria [12,14]. These maybe the reason why some bacterial taxa were more abundant in eutrophic Lake South than those in oligotrophic Lake Basomtso. However, at the same time, the growth and decomposition of harmful algae (such as Microcystis aeruginosa) could release algal toxin, which could lead to the death of some bacterial species [19]. These maybe the reason why the bacterial diversity in eutrophic lake was lower than that in oligotrophic lake. ”
Remove the first two sentences – they are in the “Introduction” section.
Response:
Thank you very much for your suggestions. As you suggested, we have removed “The lakes could be classified into three nutritional statuses: oligotrophic, mesotrophic and eutrophic lakes. It was reported that Basomtso Lake was classified as the oligotrophic lakes, while South Lake was classified as a typical eutrophication lake, which might be due to the input of nutrients such as nitrogen and phosphorus”.
Line 326: change “flora” to “group”
Response:
Thank you very much for your suggestions. As you suggested, we have corrected “flora” to “group”. (Line 340)
Line 329: “abundantly distributed” - re-word phrase.
Response:
Thank for your help. It’s my mistake, we have corrected “abundantly distributed” to “mainly distributed”. (Line 345)
Line 331: “oligotrophic groups” – unclear, maybe you mean “the α-proteobacteria are characteristic of oligotrophic lakes”
Response:
Thank you very much for your help. We have revised as “The previous study demonstrated that α-proteobacteria was the characteristic of oligotrophic lakes in aquatic environments [33,34]. Similarly, we also found oligotrophic Lake Basomtso was dominated by α-proteobacteria in sediment samples.”. (Line 341-343)
Line 335: change “Actinobacteria is” to “Actinobacteria are”
Response:
Thank you very much for your suggestions. As you suggested, we have corrected “Actinobacteria is” to “Actinobacteria are”. (Line 347)
Line 338: “eutrophication lakes” – maybe you mean “eutrophic lakes”
Response:
Thank you very much for your comments. It’s my mistake, we have corrected “eutrophication lakes” to “eutrophic lakes”. (Line 353)
Lines 340, 349, 353, 356: “more abundantly detected”, “poorly detected”, “highly detected” – incorrect phrases.
Response:
Thank you very much for your comments. As you suggested, we have corrected these incorrect phrase in the revised manuscript. (Line 357&365&379)
Lines 345-348: re-word this sentence and divide into several ones.
Thank you very much for your suggestion. We have re-word this sentence. (Line 357-360).
“In this study, the abundance of Bacteroidetes in the oligotrophic lake were higher than that in the eutrophic lake. In addition, Bacteroidetes were also more abundant in the water than that in the sediments. Similar results have also been reported in previous study [5].”
Lines 360-361: remove this sentence.
Response:
Thank you very much for your suggestions. As you suggested, we have removed “In summary, the bacterial community were examined in the sediment and water samples derived from oligotrophic Basomtso Lake and eutrophic South Lake, respectively”.
Lines 375-377 (Author Contributions) – I did not find the initials LXJ and LDP; I suppose that HM and DC should be replaced by MH and CD.
Response:
Thank you very much for your suggestions. It’s my mistake, we have corrected “XL and DL conceived the study. CF, JJ, BH and CW performed the experiments. JJ, CW, MH and CD contributed to the sample collection”. (Line 385-377)
References
Lines 437-439: “Pseudoalteromonas, Proteobacteria, Chattonella, Gymnodinium, Heterosigma” should be in italics.
Response:
Thank you very much for your suggestions. we have removed this reference.
Figure 1. caption: change “sampling” to “studied”
Response:
Thank you very much for your suggestions. As you suggested, we have corrected Figure 1. caption “sampling” to “studied”. (Line 94)
Figs 3, 4, 5 captions: change “Other” to “Others”, What is meant by the phrases “the South Lake samples have a positive difference”, “a negative difference”? I have never seen it before.
Response:
Thank you very much for your suggestions. As you suggested, we have corrected “Other” to “Others”. (Line 225, Figure 3; Line 256, Figure 4)
Sorry for the confusing. We have rephrased this sentence in Figure 3,4,5. (Line 227-229, 257-259, 280-281)
“the circle value represented the differences between the proportions of Lake Basomtso (red color) and Lake South (blue color).” (Line 227-229, 257-259)
“The circle value represented the differences between the proportions of water sample (red color) and sediment sample (blue color) in Lake Basomtso.” (Line 280-281)
Tables 1 and 2 should be combined into one table to better visualize the differences in the species composition between the studied lakes (see below). The same applies to Tables 3 and 4. Moreover, I think that Family, Order, Genus are unnecessary; in the case of zooplankton, give the species name, e.g. Brachionus sp.
|
Species |
Oligotrophic Lake Basomtso |
Eutrophic Lake South |
|
Bacillariophyta |
|
|
|
Cyclotella meneghiniana |
X |
X |
|
Cyclotella aslerocastata |
X |
|
|
|
|
|
|
Total species number |
27 |
86 |
Response:
Thank you very much for your smart suggestion. As you suggested, we have combined Supplemental Table 1 and Supplemental 2 into the new Supplemental Table1. In addition, the Supplemental Table 3 and Supplemental Table 4 were also combined into one new Supplemental Table 2.
Table 1: Oscillatoriales are not a genus, but order
Response:
Sorry for our shortcoming. We have corrected this mistake in the new Supplemental Table 1.
Table 2: “Asterionella”, “Trachelomonas” instead of “Asterionelle”, “Tiachelomonas”
Response:
Thank you very much for your careful comments. Sorry for our shortcoming. We have corrected these mistake in the new Supplemental Table 1 with yellow highlight.
Table 3: Diaptomidae are not a genus; if some individuals have not been identified, give sp.1, sp. 2, ...
Response:
Sorry for our shortcoming. We have corrected this mistake in the new Supplemental Table 2.
Reviewer 2 Report
The manuscript by Feng et al. analyzes bacterial and phytoplankton diversity between two freshwater lakes in water samples and bacterial community alone in the sediments. They observe a negative correlation between bacterial and phytoplankton in the studied freshwater systems. However they don’t go further and discuss the importance of their findings and ecological significance of it. To me it occurs that this manuscript is more of a comparison of diversity between the two lakes at a given time point and doesn’t go beyond further. Since lake ecosystems are characterized by seasonal fluctuations, it would have been more interesting to see bacterial diversity patterns with seasons and their interaction with phytoplankton. But instead they chose to restrict their study to one month (August). The authors have done a fairly decent job in analyzing bacterial community and diversity, but have not sufficiently highlighted their interaction with phytoplankton. Results and discussion pertaining to the diversity estimates between the water samples stand out separately as they fail to find a suitable link between them. The manuscript is let down by its writing. English grammar falters throughout the manuscript. Instead of pin pointing them, I would rather like the authors to take assistance from a native English speaker in checking the language.
Specific comments
L16: should be ‘play’
L18: indicate that lakes are situated in China (in brackets)
L20, 23,25 &26: dominated should be indicate by percentage value
L27: significant differences or higher? Need to give P value, elsewhere in the abstract also
L32: Need to highlight the importance of the findings and their ecological implications. Take home message.
L37: could be modified as …oligotrophic, mesotrophic and eutrophic lakes…
L38: could be modified as … excessive input of nutrients, mainly by nitrogen and phosphorous.
L40: should be… changes in nutrient concentration could affect…
L42: Modify this paragraph. Among microorganisms, heterotrophic bacteria play a viral…….
L57-71: Need to highlight the importance of phyto-bacteria relationship in lakes and need reasons why sediments were included in your study. Main objectives should be clearly defined. Hypothesis need to be added.
L68-70: Need to be rephrased.
L75-76: What are the depths of the lakes?
L84: Provide reference for HL-CN mud collector. In what time the samples were transported to the laboratory.
L85-88: How much ml of samples was filtered? What type of filters was used?
Methodology on phytoplabkton diversity is missing
L94: change it to ‘Bacterial DNA’…..
In the methods introduce a section on “statistical analyses” Throughout the manuscript the authors refer to as higher than or lower than when comparing their data. This should be supported by statistical analyses. If it is significant the authors need to show the p value.
L156-158: It is quite clear that this study involved 2 lakes. No need to repeat it often.
L185: are showed in Fig.3
L189: change it to ‘diversity’
L193: significantly higher (p value). Pls check elsewhere also.
L315: Similar to first line of introduction.
L345-348: sentence too long.
Finally the authors should bring out the phyto-bacteria relationship and their ecological significance of these two contrasting lake systems. Clearly the data points are not enough, and should include more sampling months representative of seasons. This could eventually bring out the existing relationship of phyto and bacteria with time.
Author Response
Reviewer 2
Comments and Suggestions for Authors
The manuscript by Feng et al. analyzes bacterial and phytoplankton diversity between two freshwater lakes in water samples and bacterial community alone in the sediments. They observe a negative correlation between bacterial and phytoplankton in the studied freshwater systems. However, they don’t go further and discuss the importance of their findings and ecological significance of it. To me it occurs that this manuscript is more of a comparison of diversity between the two lakes at a given time point and doesn’t go beyond further. Since lake ecosystems are characterized by seasonal fluctuations, it would have been more interesting to see bacterial diversity patterns with seasons and their interaction with phytoplankton. But instead they chose to restrict their study to one month (August). The authors have done a fairly decent job in analyzing bacterial community and diversity, but have not sufficiently highlighted their interaction with phytoplankton. Results and discussion pertaining to the diversity estimates between the water samples stand out separately as they fail to find a suitable link between them. The manuscript is let down by its writing. English grammar falters throughout the manuscript. Instead of pin pointing them, I would rather like the authors to take assistance from a native English speaker in checking the language.
Response:
Thank you very much for your kind comments and suggestions on our manuscript. We also realized the shortcoming on our manuscript. As you suggested, we have tried our best to revise the manuscript again.
Q (1) The manuscript by Feng et al. analyzes bacterial and phytoplankton diversity between two freshwater lakes in water samples and bacterial community alone in the sediments. They observe a negative correlation between bacterial and phytoplankton in the studied freshwater systems. However, they don’t go further and discuss the importance of their findings and ecological significance of it. To me it occurs that this manuscript is more of a comparison of diversity between the two lakes at a given time point and doesn’t go beyond further. Since lake ecosystems are characterized by seasonal fluctuations, it would have been more interesting to see bacterial diversity patterns with seasons and their interaction with phytoplankton. But instead they chose to restrict their study to one month (August).
Response:
(1) Thank you very much for your help with the paper review. In this study, to clarify the phytoplankton and bacterial community structure in two different trophic lakes, the oligotrophic Lake Basomtso and eutrophic Lake South were used as the models. Firstly, the phytoplankton community composition in the water samples from the two lakes were examined. Secondly, the bacterial community in the water and sediment from the two lakes were analyzed by using 16S rRNA sequences. Finally, to determine whether bacteria show different assembly patterns in different habits, the present study also investigated the bacterial abundance and community composition in water and sediment of oligotrophic lake. This study will contribute to better understand the interactions between bacteria and phytoplankton in different trophic lakes.
(2) You are right that our one month’s dataset (three data for surface waters and sediments, and two data for bottom waters for each lake) could not support the conclusion that “there was a negative correlation between the phytoplankton diversity and bacterial diversity in freshwater ecosystem”. As you suggested, we have removed this conclusion and revised our abstract again. (Line 15-31)
“Abstract: Phytoplankton are the primary producers at the basis of aquatic food webs, and bacteria play an important role in energy flow and biochemical cycling in aquatic ecosystems. In this study, both the bacterial and phytoplankton communities were examined in the oligotrophic Lake Basomtso (China) and the eutrophic Lake South (China). Microscopic examination showed that the phytoplankton density and diversity in eutrophic lake were higher than those in oligotrophic lake. Furthermore, Chlorophyta (68.26%) and Cryptophyta (24.48%) were the dominant groups in eutrophic lake, while Bacillariophyta (94.96%) were dominated in oligotrophic lake. The bacterial communities in the waters and sediments of the two lakes were mainly composed of Proteobacteria (mean of 32.05%), Actinobacteria (mean of 24.64%), Bacteroidetes (mean of 11.93%), and Chloroflexi (mean of 6.12%). Comparative analysis showed that the abundance of bacteria in eutrophic lake were higher than that in oligotrophic lake (P<0.05), but the bacterial diversity in oligotrophic lake was higher than that in eutrophic lake (P<0.05). Finally, the bacterial abundance and diversity in the sediments of the two lakes were higher than those in the water samples (P<0.05), and the Latescibacteria and Nitrospinae groups were identified only in the sediments. These results, taken together, suggested that both phytoplankton and bacterial community were different in the oligotrophic lake and eutrophic lake.”
(3) We also thank a lot for your good suggestion. As you suggested, in our further study, we will choose three different trophic lakes to sample monthly January-December to determine the correlations between the phytoplankton and bacteria in different trophic lakes.
Q (2) The authors have done a fairly decent job in analyzing bacterial community and diversity, but have not sufficiently highlighted their interaction with phytoplankton. Results and discussion pertaining to the diversity estimates between the water samples stand out separately as they fail to find a suitable link between them.
Response:
Thank you very much for your comments. As you suggested, we have added more information about the interaction between phytoplankton and bacteria in introduction (Line 45-52) and discussion (Line 324-334).
Q (3) The manuscript is let down by its writing. English grammar falters throughout the manuscript. Instead of pin pointing them, I would rather like the authors to take assistance from a native English speaker in checking the language.
Response:
Thank you very much for your comments. Sorry for our shortcoming in the English. As you suggested, we have tried our best to revise the manuscript again. In addition, we also take assistance from one English teacher in our school to check the language.
Specific comments
L16: should be ‘play’
Response:
Thank you very much for your suggestions. As you suggested, we have corrected “played” to “play”. (Line 16)
L18: indicate that lakes are situated in China (in brackets)
Response:
Thank you very much for your suggestions. As you suggested, we have marked China behind the lakes. (Line 18)
L20, 23,25 &26: dominated should be indicate by percentage value
Response:
Thank you very much for your suggestions. As you suggested, the percentage value has been added in the revised abstract. (Line 20,23 &24)
L27: significant differences or higher? Need to give P value, elsewhere in the abstract also
Response:
Thank you very much for your suggestions. It is “significant differences”. As you suggested, we have added the P value in the revised manuscript. (Line 25, 27, 151, 153 &156)
L32: Need to highlight the importance of the findings and their ecological implications. Take home message.
Response:
Thank you very much for your suggestion. According to our results, we have revised our conclusion in the abstract. (Line 28-30)
L37: could be modified as …oligotrophic, mesotrophic and eutrophic lakes…
Response:
Thank you very much for your suggestions. As you suggested, we have corrected “oligotrophic lake, mesotrophic lake and eutrophic lake” to “oligotrophic, mesotrophic and eutrophic lakes”. (Line 35)
L38: could be modified as … excessive input of nutrients, mainly by nitrogen and phosphorous.
Response:
Thank you very much for your suggestions. As you suggested, we have corrected “excessive nutrient content such as nitrogen and phosphorus” to “excessive input of nutrients, mainly by nitrogen and phosphorous”. (Line 36)
L40: should be… changes in nutrient concentration could affect…
Response:
Thank you very much for your suggestions. As you suggested, we have corrected “changes of nutrients could affect” to “changes in nutrient concentration could affect”. (Line 38)
L42: Modify this paragraph. Among microorganisms, heterotrophic bacteria play a viral…….
Response:
Thank you very much for your comments. As you suggested, we have rephrased this paragraph. (Line 40-45)
“Microorganisms were both the producers and decomposers of organic matters in aquatic ecosystems, which play an important role in regulating the circulation of biogenic elements such as carbon, nitrogen, phosphorus and sulfur in lakes [6,7]. Bacteria, as the important part of the microbial community, are mainly responsible for the mineralization of organic matter and the recycling, while the recycling of dissolved organic carbon in lake water were mainly fulfilled by heterotrophic bacteria [8-10].”
L57-71: Need to highlight the importance of phyto-bacteria relationship in lakes and need reasons why sediments were included in your study. Main objectives should be clearly defined. Hypothesis need to be added.
Response:
Thank you very much for your constructive suggestions. As you suggested, we have added phyto-bacteria relationship in our introduction. (Line 45-52)
“Phytoplankton are the primary producers at the basis of aquatic food webs, which can promptly respond to environmental changes [11]. In aquatic ecosystems, interactions between phytoplankton and bacteria have been proposed to influence bacterial community [12-14]. Phytoplankton could provide shelter for predation-related bacteria, as well as the habitat for endophytic bacteria living within the algae cells and epiphytic bacteria on the surface of algae [15,16]. In addition to providing the shelter, phytoplankton could provide organic matter (e.g. sugars, amino acids) to bacteria [12]. Interestingly, phytoplankton could also display a negative affect in bacterial community through nutrient competition and antibiotic release [17].”
Thank you very much for your constructive suggestions. As you suggested, we have revised this paragraph for the objectives in our study. (Line 61-68)
“In this study, to clarify the phytoplankton and bacterial community structure in two different trophic lakes, the oligotrophic Lake Basomtso and eutrophic Lake South were used as the models. Firstly, the phytoplankton community composition in the water samples from the two lakes were examined. Secondly, the bacterial community in the water and sediment from the two lakes were analyzed by using 16S rRNA sequences. Finally, to determine whether bacteria show different assembly patterns in different habits, the present study also investigated the bacterial abundance and community composition in water and sediment of oligotrophic lake. This study will contribute to better understand the interactions between bacteria and phytoplankton in different trophic lakes.”
L68-70: Need to be rephrased.
Response:
Thank you very for your suggestion. As you suggested, we have rephrased this sentence. (Line 67-68)
L75-76: What are the depths of the lakes?
Response:
Thank you very much for your comments. As you suggested, we have added morphometric characteristics (area, and depth) of lakes it in M&M. (Line 71-75)
“Lake Basomtso is one of the largest freshwater dammed lake in the east of Tibet, and is a oligotrophic freshwater lake. Its length, area and average depth are 18 km, 27 km2 and 68 m, respectively (Fig. 1). Lake South is a typical eutrophic lake in Wuhan City, which is located in the Middle-Lower Yangtze plains. Its area and average depth are 5.50 km2 and 1.6 m, respectively (Fig. 1).”
L84: Provide reference for HL-CN mud collector. In what time the samples were transported to the laboratory.
Response:
Thank you very much for your comments. Sorry for the confusing. All the samples stored at 4°C incubator was transported to the laboratory within two hours. Then the sediment samples were stored in -80°C until further procedures. (Line 86-88)
L85-88: How much ml of samples was filtered? What type of filters was used?
Response:
Thank you very much for your comments. As you suggested, the information for the filters has been added in the revised manuscript. (Line88-91 )
“The water samples for high-throughput sequencing were filtered through 5.0 μm pore size membrane filters (Millipore, Ireland) using a diaphragm vacuum pump (JinTeng GM-0.5B, Tianjin, China) to remove particle impurities, and then 1000 mL were collected on 0.22 μm pore size membrane filters (Millipore, Ireland).” (Line 88-91)
Methodology on phytoplabkton diversity is missing
Response:
Thank you very much for your comments. Sorry for our shortcoming. As you suggested, the methods for plankton diversity have been added in the Materials and methods. (Line 95-106)
“2.2 Qualitative and quantitative analyses of plankton
The phytoplankton samples were collected by using 1L sampler, and the samples were fixed in Lugol’s solution. Then, the fixed samples were transported to the laboratory and precipitated by DXCJQ-1L plankton settler within 48 h. Finally, 50 mL concentrated sample was secondary fixed by using 1mL formaldehyde. The zooplankton samples were collected by using the 5 L sampler. Then, the samples were filtered through the 13# plankton net and fixed in formaldehyde solution. Using a 10 mL sedimentation chamber, the qualitative and quantitative analysis of plankton in oligotrophic lakes was performed by inverted microscope technique with 100× and 400× magnifications, respectively. The phytoplankton and zooplankton samples in the eutrophic lake were qualitatively and quantitatively counted by an Optical Microscope using 0.1 mL and 1 mL counting chamber, respectively. Besides, species identification was performed as previous studies [24-26].”
L94: change it to ‘Bacterial DNA’….
Response:
Thank you very much for your suggestions. As you suggested, we have corrected “Microbial DNA” to “Bacterial DNA”. (Line 108)
In the methods introduce a section on “statistical analyses” Throughout the manuscript the authors refer to as higher than or lower than when comparing their data. This should be supported by statistical analyses. If it is significant the authors need to show the p value.
Response:
Thank you very much for your comments. As you suggested, the section on statistical analyses has been added in the Material and Methods. (Line 140-144)
“The three replicates were expressed as Mean ± SD; all data were tested for normality of distribution using the Shapiro-Wilk normality test. Then, one-way ANOVA was used to test the significant difference according to different experiment groups. The differences between groups were considered as significant at P<0.05 (“*”), highly significant at P<0.01 (“**”) or at P<0.001 (“***”).”
L156-158: It is quite clear that this study involved 2 lakes. No need to repeat it often.
Response:
Thank you very much for your suggestions. We have removed “High-throughput sequencing technique was used to perform the bacterial community diversity in the water and sediment samples from two different trophic lakes, Basomtso Lake (Tibet, China) and South Lake (Wuhan, China), respectively”.
L185: are showed in Fig.3
Response:
Thank you very much for your suggestions. As you suggested, we have corrected “were showed in Fig. 3” to “are showed in Fig.3”. (Line 204)
L189: change it to ‘diversity’
Response:
Thank you very much for your suggestions. As you suggested, we have corrected “diversities” to “diversity”. (Line 208)
L193: significantly higher (p value). Pls check elsewhere also.
Response:
Thank you very much for your comments. As you suggested, the p value has been added in the revised manuscript. (Line 209, 151, 152, 154,155)
L315: Similar to first line of introduction.
Response:
Thank you very much for your suggestions. We have removed “The lakes could be classified into three nutritional statuses: oligotrophic, mesotrophic and eutrophic lakes. It was reported that Basomtso Lake was classified as the oligotrophic lakes, while South Lake was classified as a typical eutrophication lake, which might be due to the input of nutrients such as nitrogen and phosphorus.”
L345-348: sentence too long.
Response:
Sorry for our shortcoming. As you suggested, the sentence has been divided into two parts. (Line 358-360)
“In this study, the abundance of Bacteroidetes in the oligotrophic lake were higher than that in the eutrophic lake. In addition, Bacteroidetes were also more abundant in the water than that in the sediments. Similar results have also been reported in previous study [5].”
Finally, the authors should bring out the phyto-bacteria relationship and their ecological significance of these two contrasting lake systems. Clearly the data points are not enough, and should include more sampling months representative of seasons. This could eventually bring out the existing relationship of phyto and bacteria with time.
Response:
Thank you very much for your constructive suggestions.
You are right that our one month’s dataset (three data for surface waters and sediments, and two data for bottom waters for each lake) could not support the conclusion that “there was a negative correlation between the phytoplankton diversity and bacterial diversity in freshwater ecosystem”. As you suggested, we have revised our conclusion in the abstract. (Line 26-30)
We also thank a lot for your constructive suggestion. Following your suggestion, in our further study, we will choose three different trophic lakes to sample monthly January-December to further determine the correlations between the phytoplankton and bacteria in different trophic lakes.
Reviewer 3 Report
The manuscript is devoted to the study of the characteristics of the bacterial community in the oligotrophic and eutrophic lakes. The authors described bacteria in detail both in water and in sediments. This study will give a significant contribution to enriching our information for the bacterial communities in the freshwater ecosystem.
However, I did not find research methods and results on phytoplankton. Accordingly, the conclusion about the diversity of phytoplankton is not substantiated. I suggest that the authors remove phytoplankton from the title of the manuscript and from the text. Supplimentary materials contain scarce information about phytoplankton and zooplankton community. These data are not discussed in any way in the manuscript. Therefore, I also propose to remove.
Author Response
Comments and Suggestions for Authors
The manuscript is devoted to the study of the characteristics of the bacterial community in the oligotrophic and eutrophic lakes. The authors described bacteria in detail both in water and in sediments. This study will give a significant contribution to enriching our information for the bacterial communities in the freshwater ecosystem.
However, I did not find research methods and results on phytoplankton. Accordingly, the conclusion about the diversity of phytoplankton is not substantiated. I suggest that the authors remove phytoplankton from the title of the manuscript and from the text. Supplimentary materials contain scarce information about phytoplankton and zooplankton community. These data are not discussed in any way in the manuscript. Therefore, I also propose to remove.
Response:
Thank you very much for your help with the paper review. We also realized the shortcoming on our manuscript. As you suggested, we have tried our best to revise the manuscript again.
However, I did not find research methods and results on phytoplankton.
Response:
Thank you very much for your comments. Sorry for our shortcoming. As you suggested, we have added the methods for plankton collection and analysis in the revised manuscript. (Line 95-106)
“2.2 Qualitative and quantitative analyses of plankton
The phytoplankton samples were collected by using 1L sampler, and the samples were fixed in Lugol’s solution. Then, the fixed samples were transported to the laboratory and precipitated by DXCJQ-1L plankton settler within 48 h. Finally, 50 mL concentrated sample was secondary fixed by using 1mL formaldehyde. The zooplankton samples were collected by using the 5 L sampler. Then, the samples were filtered through the 13# plankton net and fixed in formaldehyde solution. Using a 10 mL sedimentation chamber, the qualitative and quantitative analysis of plankton in oligotrophic lakes was performed by inverted microscope technique with 100× and 400× magnifications, respectively. The phytoplankton and zooplankton samples in the eutrophic lake were qualitatively and quantitatively counted by an Optical Microscope using 0.1 mL and 1 mL counting chamber, respectively. Besides, species identification was performed as previous studies [24-26].”
Accordingly, the conclusion about the diversity of phytoplankton is not substantiated. I suggest that the authors remove phytoplankton from the title of the manuscript and from the text.
Response:
Thank you very much for your suggestions. In this study, we want to examine the interaction between bacteria and phytoplankton in freshwater lakes, so we would like to keep the phytoplankton in the manuscript. As you suggested, we have added more information about the phytoplankton in the introduction (Line 45-52), results (Line 162-178, Table 1), and discussion (Line 324-334).
Introduction (Line 45-52)
“Phytoplankton are the primary producers at the basis of aquatic food webs, which can promptly respond to environmental changes [11]. In aquatic ecosystems, interactions between phytoplankton and bacteria have been proposed to influence bacterial community [12-14]. Phytoplankton could provide shelter for predation-related bacteria, as well as the habitat for endophytic bacteria living within the algae cells and epiphytic bacteria on the surface of algae [15,16]. In addition to providing the shelter, phytoplankton could provide organic matter (e.g. sugars, amino acids) to bacteria [12]. Interestingly, phytoplankton could also display a negative affect in bacterial community through nutrient competition and antibiotic release [17].”
Results (Line 162-178)
“3.2. Plankton community composition in the two lakes
Microscopic examination showed that the total phytoplankton density in eutrophic Lake South (4.82×108 cells/L) was higher than that in oligotrophic Lake Basomtso (3933 cells /L) (Table 1). Among them, Bacillariophyta accounted for 94.96% of total phytoplankton density in oligotrophic lake, whereas the Chlorophyta and Cyanophyta accounted for 68.26% and 24.48% of total phytoplankton density in eutrophic Lake South, respectively. In addition, there were 22 genera, 16 families and 6 phylum phytoplankton in Lake Basomtso, while the phytoplankton in Lake South was composed of 6 phylums, 23 families and 36 genera (Table S1). Among them, the main dominant species were Cyclotella meneghiniana, Nitzschia linearis and Surirella biseriate in oligotrophic Lake Basomtso (Table S1). In contrast, the main dominant species in Lake South were Anabaena circinalis, Anabaena viguieri, Microcystis wesenbergii, and Merismopedia minima (Table S1).
Several zooplankton were detected in Lake Basomtso, including rotifers (4 families and 6 genera) and copepods (3 families and 3 genera) (Table S2). Among them, Diaptomidae were dominanted in Bsomtso Lake (Table S2). In addition, compared to Lake Basomtso, the zooplankton in Lake South were more abundant, including rotifers (5 families and 6 genera), copepods (1 family and 1 genera), and cladocerans (1 family and 1 genera) (Table S2). Among them, the rotifers were the most abundant community in Lake South (Table S2).”
Discussion (Line 324-334)
“In the present study, both the nutrient (TP, TN) and temperature in eutrophic Lake South were higher than those in oligotrophic Lake Basomtso. As we know, the high excessive nutrient and temperature could induce the growth and reproduction of planktonic algae and significantly increase the biomass of phytoplankton [27]. These maybe the reason why the phytoplankton density in eutrophic South Laker were higher than that in oligotrophic Lake Basomtso. Furthermore, the growth and decomposition of algae will not only increase the pH of water, but also induce the growth of bacteria [12,14]. These maybe the reason why some bacterial taxa were more abundant in eutrophic Lake South than those in oligotrophic Lake Basomtso. However, at the same time, the growth and decomposition of harmful algae (such as Microcystis aeruginosa) could release algal toxin, which could lead to the death of some bacterial species [19]. These maybe the reason why the bacterial diversity in eutrophic lake was lower than that in oligotrophic lake.”
Supplementary materials contain scarce information about phytoplankton and zooplankton community. These data are not discussed in any way in the manuscript. Therefore, I also propose to remove.
Response:
Thank you very much for your comments. Sorry for our shortcoming. As you suggested, we have revised the Supplemental tables (Supplemental Table 1, Supplemental Table 2). In addition, we also discussed this results in the revised discussion. (Line 324-3334)
Round 2
Reviewer 1 Report
Compared to the first submission, the manuscript gained a little in clarity and precision, but not enough. Although the Introduction and Discussion sections have been developed slightly, they still require further consideration (see comments below). Some parts of the manuscript are still difficult to read and understand. For this reasons, I suggest the publication of this manuscript in Microorganisms after a satisfactory revision. First of all, the manuscript needs a revision by a native English speaker!
Specific comments
Abstract
Line 18: change “Lake Basomtso (China) and the eutrophic Lake South (China).” to “Lake Basomtso and the eutrophic Lake South (China).” Change “Microscopic examination showed” to “The results of this study showed”
Lines 20, 21, 23, 24: percent values are given in great detail; 68%, 24%, 95%, … are enough.
Line 21: change “were dominated” to “dominated”
Line 29: delete “taken together” and change “suggested” to “suggest”, change “were different in” to “differed considerably between”
Introduction
I think that this section has not been developed sufficiently. What methods were applied to estimate taxonomic composition of bacterial communities? (you should mention various techniques, e.g. FISH, CARD-FISH, DGGE,16S rRNA genes), what it is known about bacterial (and phytoplankton) taxonomic structure in lakes of different trophic status? what are the reasons for serious changes in bacterial community composition? – an overview should be added here. There are still no hypotheses to be tested.
Line 39: what is meant by the phrase “nutritional status”?
Line 46: change “promptly” to “quickly” or “rapidly”
Lines 50-51: this sentence is unclear; “for bacteria” instead of “to bacteria”.
Line 52: “affect in bacterial community” – should be (probably) “effect (or impact, influence) on bacterial community”.
Materials and Methods
Lines 71-80: this part of the Study site and sampling should be corrected. I suggest: The study was conducted in oligotrophic Lake Basomtso and eutrophic Lake South. Lake Basomtso (area of 27 km2, maximum depth of…, and mean depth of 68 m) is one of the largest freshwater dammed lake in the east of Tibet. Lake South (area of 5.50 km2, maximum depth of…, and mean depth of 1.6 m) is located in Wuhan City in the Middle-Lower Yangtze plains. In both lakes, samples were collected in August 2018. In oligotrophic Lake Basomtso, samples were collected from the surface (O-S1, O-S2, O-S3) and bottom water layers (O-B1, O-B2) as well as from the sediments (O-SE1, O-SE2, O-SE3) from three sites situated from river inlet to lake center. In eutrophic Lake South, samples were collected from the surface water layers (E-S1, E-S2, E-S3) and from the sediments (E-SE1, E-SE2) from three sites situated from the shore to the center. Location of the studied lakes and sampling sites are presented in Fig. 1.” In addition, sampling sites should be explained in figures captions, for instance, Figure 2. (A) Rarefaction curves base on high-throughput sequencing. (B) Venn diagrams of shared OTUs. O-S - oligotrophic lake, surface waters, O-B - oligotrophic lake, bottom waters, O-SE - oligotrophic lake, sediments, E-S - eutrophic lake, surface waters, E-SE - eutrophic lake, sediments.
Lines 96, 99: change “were collected by using” to “were collected using a”.
Line 97: change “fixed in Lugol’ s solution” to “fixed with Lugol’s solution”.
Lines 98-99: change “fixed by using 1mL formaldehyde” to “fixed with 1mL formaldehyde”.
Line 100: delete “13#”.
Line 146: the title of this sub-chapter should be changed. I suggest, for instance: “Environmental conditions”
Results and Discussion
In general, the authors should make more effort to better describe and discuss the results.
Line 147-149: delete this sentence.
Line 151: change “the water temperature in Lake Basomtso were highly significant lower than that in Lake South (P<0.001).” to “water temperature was significantly lower in Lake Basomtso than in Lake South (P<0.001)”.
Line 163: delete “Microscopic examination showed that”
Lines 167-169: correct this sentence according to the information presented in Table 1 (in this version of the manuscript, there are phylum and genus/species only; the total number of species is important!). Similarly, please correct data for rotifers and crustaceans. The results concerning zooplankton community com position seem to be incomplete (surprisingly low number of species, lack of the total numbers and/or biomass of both rotifers and crustaceans). That is why, they are unnecessary in my opinion. Moreover, these zooplankton data did not refer to bacterial and phytoplankton abundance and composition in the Discussion section (also in the Introduction); the title of the manuscript does not include zooplankton as well.
Discussion: The whole discussion is like a general review and it is based only on the comparison of the obtained results with those of other studies. The discussion should focus on the novelty of the findings obtained in this study. It is hard to know which groups of bacteria dominated both in the studied lakes and other freshwater ecosystems (e.g. lines 338-341: “Several previous studies have reported that the bacterial communities in the freshwater lakes were mainly dominated by Proteobacteria [3,5,22,28]. Proteobacteria was also the most dominant group in both Lake South and Lake Basomtso. In addition, previous studies have showed that the β-proteobacteria group was mainly dominated in the freshwater microbial community [29,30], which was also consistent with the results in the present study” and then line 348: “Actinobacteria are the most dominant bacterial group in freshwater ecosystems [36].”).
Table 1: “sp.” and “nauplii” should not be in italics.
Change “emergence” to “presence”
Change “Pseudoanabnena sp”.to “Pseudanabaena sp.”
Diatoma vulgare - accepted name is Diatoma vulgaris var. vulgaris Bory de Saint-Vincent, 1824.
Change „Khawkinea acutecouato” to „Khawkinea acutecaudata”; in the previous version of the manuscript, this species was not mentioned. Moreover, in the previous version, the total number of species in Lake South was 86, but in the revised version it is 97 species – why? Please check carefully other genus/species names!
Author Response
Compared to the first submission, the manuscript gained a little in clarity and precision, but not enough. Although the Introduction and Discussion sections have been developed slightly, they still require further consideration (see comments below). Some parts of the manuscript are still difficult to read and understand. For this reasons, I suggest the publication of this manuscript in Microorganisms after a satisfactory revision. First of all, the manuscript needs a revision by a native English speaker!
Response:
Thank you very much for your help with the paper review. As your kind suggestions, our manuscript has been improved again. Firstly, the “Introduction” and “Discussion” have been rewritten to make them more logical. Secondly, the manuscript has been revised with proper editing to remove/minimize the errors for grammar.
Specific comments
Abstract
Line 18: change “Lake Basomtso (China) and the eutrophic Lake South (China).” to “Lake Basomtso and the eutrophic Lake South (China).” Change “Microscopic examination showed” to “The results of this study showed”
Response:
Thank you very much for your suggestions. As you suggested, we have changed “Lake Basomtso (China) and the eutrophic Lake South (China).” to “Lake Basomtso and the eutrophic Lake South (China).” Line 18. Changed “Microscopic examination showed” to “The results of this study showed”. (Line 18)
Lines 20, 21, 23, 24: percent values are given in great detail; 68%, 24%, 95%, … are enough.
Response:
Thank you very much for your comments. As you suggested, the percentage values have been revised as 68%, 24% (Line 20), 95% (Line 21), 32% (Line 22), 25%, 12%, 6% (Line 23).
Line 21: change “were dominated” to “dominated”
Response:
Thank you very much for your suggestions. As you suggested, we have corrected “were dominated” to “dominated”. (Line 21)
Line 29: delete “taken together” and change “suggested” to “suggest”, change “were different in” to “differed considerably between”
Response:
Thank you very much for your suggestions. As you suggested, we have removed “taken together” (Line 28); corrected “suggested” to “suggest” (Line 28); “were different in” to “differed considerably between”. (Line 29)
Introduction
I think that this section has not been developed sufficiently. What methods were applied to estimate taxonomic composition of bacterial communities? (you should mention various techniques, e.g. FISH, CARD-FISH, DGGE,16S rRNA genes), what it is known about bacterial (and phytoplankton) taxonomic structure in lakes of different trophic status? what are the reasons for serious changes in bacterial community composition? – an overview should be added here. There are still no hypotheses to be tested.
Responses:
Thank you very much for your comments. As you suggested, the introduction has been revised and added more information.
In the first paragraph, the functional role of bacteria and phytoplankton in aquatic ecosystem were introduced. In addition, we also introduced the relationship between bacteria and phytoplankton in aquatic ecosystem. (Line 34-47)
“Microorganisms are both the producers and decomposers of organic matters in aquatic ecosystems, which play an important role in regulating the circulation of biogenic elements such as carbon, nitrogen, phosphorus and sulfur in lakes [1,2]. Bacteria, as an important part of the microbial community, are mainly responsible for the mineralization of organic matter and the recycling, while the recycling of dissolved organic carbon (DOC) in lake were mainly fulfilled by heterotrophic bacteria [3-5]. Phytoplankton are the primary producers at the basis of aquatic food webs, which can quickly respond to environmental changes [6]. In lakes, interactions between phytoplankton and bacteria have been proposed to influence bacterial community dynamics [7-9]. Bacteria rapidly utilize exudates released by phytoplankton (e.g. sugars, amino acids), as well as detritus following algal cell death [7][10]. In addition to providing a source of organic matter, phytoplankton can provide a habitat to endophytic bacteria living within algae cells and epiphytic bacteria that live in the phycosphere surrounding algal cells [11,12]. Meanwhile, phytoplankton could also display a negative effect on bacterial community through nutrient competition and antibiotic release [13].”
In the second paragraph, three points were introduced. (1) what it is known about bacterial taxonomic structure in lakes of different trophic status? (2) what are the reasons for serious changes in bacterial community composition? (3) Why do we need to clarify the bacterial community structure in two different trophic lakes? (our hypotheses) (Line 48-59)
“According to biological productivity, lakes can be divided into three trophic types, including oligotrophic, mesotrophic and eutrophic lakes [14,15]. In the aquatic environment, the compositions and diversities of bacterial community may vary with the water quality. Generally, the community dynamics of aquatic bacteria vary with biotic and abiotic environmental variables, e.g. temperature, availability of nutrients, predation and interactions with other organisms, including phyto- and zooplankton [16-18]. Previous studies have showed that the main driving factors, such as nitrogen, phosphorus and temperature, could alter the taxonomical structure of bacterial community in freshwater lakes [19-21]. However, these studies were mainly focused on the taxonomical composition of sediment community [22-25]. Little is known about the bacterial community in different trophic waterbody [26]. Comparing the structure of bacterial communities of different trophic lakes could provide valuable information to protect and remediate these lakes. Hence, it is essential to explore the bacterial communities in different trophic lakes.”
In the third paragraph, we try to introduce more methods that applied to estimate taxonomic composition of bacterial communities, especially 16S rRNA sequencing technique. (Line 60-68)
“Previous studies mainly focused on the bacterial diversity in lake sediments using traditional isolation methods [27,28] and conventional DNA-based molecular methods (e.g., DGGE, T-RFLP, Q-PCR, FISH, RAPD, clone libraries) [29-34]. Recently, 16S rRNA sequencing on the Illumina MiSeq platform can provide more detailed information about microbial community diversity and structure [35-38]. This technique has been widely used in early detection of aquatic invasion and investigation of water biodiversity [38]. It could not only judge the distribution of species and analyze the community structure of species, but also effectively improve the efficiency and quality of aquatic ecosystems detection. Furthermore, it could formulate corresponding effective protection measures for maintaining ecological balance [38-40].”
In the last paragraph, we listed our objectives in the present study. (Line 69-75)
“In this study, to clarify the phytoplankton and bacterial community structure in two different trophic lakes, the oligotrophic Lake Basomtso and eutrophic Lake South were used as the models. Firstly, the phytoplankton community composition in the water samples from the two lakes were examined. Secondly, the bacterial community in the water and sediment from the two lakes were analyzed by using 16S rRNA sequences. Finally, to determine whether bacteria showed different assembly patterns in different habits, the present study also compared the bacterial community composition in water and sediment of oligotrophic lake.”
Line 39: what is meant by the phrase “nutritional status”?
Response:
Thank you very much for your comments. We have removed “nutritional status”.
Line 46: change “promptly” to “quickly” or “rapidly”
Response:
Thank you very much for your suggestions. As you suggested, we have corrected “promptly” to “quickly”. (Line 40)
Lines 50-51: this sentence is unclear; “for bacteria” instead of “to bacteria”.
Response:
Thank you very much for your suggestions. As you suggested, we have corrected this sentence. (Line 45)
Line 52: “affect in bacterial community” – should be (probably) “effect (or impact, influence) on bacterial community”.
Response:
Thank you very much for your suggestions. As you suggested, we have corrected “affect in bacterial community” to “effect on bacterial community”. (Line 46)
Materials and Methods
Lines 71-80: this part of the Study site and sampling should be corrected. I suggest: The study was conducted in oligotrophic Lake Basomtso and eutrophic Lake South. Lake Basomtso (area of 27 km2, maximum depth of…, and mean depth of 68 m) is one of the largest freshwater dammed lake in the east of Tibet. Lake South (area of 5.50 km2, maximum depth of…, and mean depth of 1.6 m) is located in Wuhan City in the Middle-Lower Yangtze plains. In both lakes, samples were collected in August 2018. In oligotrophic Lake Basomtso, samples were collected from the surface (O-S1, O-S2, O-S3) and bottom water layers (O-B1, O-B2) as well as from the sediments (O-SE1, O-SE2, O-SE3) from three sites situated from river inlet to lake center. In eutrophic Lake South, samples were collected from the surface water layers (E-S1, E-S2, E-S3) and from the sediments (E-SE1, E-SE2) from three sites situated from the shore to the center. Location of the studied lakes and sampling sites are presented in Fig. 1.” In addition, sampling sites should be explained in figures captions, for instance, Figure 2. (A) Rarefaction curves base on high-throughput sequencing. (B) Venn diagrams of shared OTUs. O-S - oligotrophic lake, surface waters, O-B - oligotrophic lake, bottom waters, O-SE - oligotrophic lake, sediments, E-S - eutrophic lake, surface waters, E-SE - eutrophic lake, sediments.
Response:
Thank you very much for your suggestions. As you suggested, we have made changed. (Line 78-87; Line 102; Line 189-191; Line 224-225; Line 256; Line 275; Line 288-2909; Line 307-309; Line 320-322)
Lines 96, 99: change “were collected by using” to “were collected using a”.
Response:
Thank you very much for your suggestions. As you suggested, we have corrected “were collected by using” to “were collected using a”. (Line 104)
Line 97: change “fixed in Lugol’ s solution” to “fixed with Lugol’s solution”.
Response:
Thank you very much for your suggestions. As you suggested, we have corrected “fixed in Lugol’ s solution” to “fixed with Lugol’s solution”. (Line 105)
Lines 98-99: change “fixed by using 1mL formaldehyde” to “fixed with 1mL formaldehyde”.
Response:
Thank you very much for your suggestions. As you suggested, we have corrected “by using 1 mL formaldehyde” to “with 1 mL formaldehyde”. (Line 106)
Line 100: delete “13#”.
Response:
Thank you very much for your suggestions. As you suggested, we have removed “13#”.
Line 146: the title of this sub-chapter should be changed. I suggest, for instance: “Environmental conditions”
Response:
Thank you very much for your suggestions. As you suggested, we have modified the title of this sub-chapter as “Environmental conditions”. (Line 152)
Results and Discussion
In general, the authors should make more effort to better describe and discuss the results.
Line 147-149: delete this sentence.
Response:
Thank you very much for your suggestions. As you suggested, we have removed “To better elucidate the relationship among the phytoplankton, bacteria and physic-chemical characteristics of the waters they inhabited, the surface hydrography and nutrient profiles were detected in eutrophic Lake South and oligotrophic Lake Basomtso during our sampling.”
Line 151: change “the water temperature in Lake Basomtso were highly significant lower than that in Lake South (P<0.001).” to “water temperature was significantly lower in Lake Basomtso than in Lake South (P<0.001)”.
Response:
Thank you very much for your suggestions. As you suggested, we have corrected this sentence. (Line 154)
Line 163: delete “Microscopic examination showed that”
Response:
Thank you very much for your suggestions. As you suggested, we have removed “Microscopic examination showed that”. (Line 166)
Lines 167-169: correct this sentence according to the information presented in Table 1 (in this version of the manuscript, there are phylum and genus/species only; the total number of species is important!).
Response:
Thank you very much for your comments. As you suggested, this sentence has been corrected according to the information presented in Supplemental Table 1. (Line 170-171)
“Additionally, numbers of phytoplankton species in oligotrophic lake (21 genera, 6 phyla) were lower than that in eutrophic lake (50 genera, 6 phyla) (Supplemental Table 1).”
Similarly, please correct data for rotifers and crustaceans. The results concerning zooplankton community composition seem to be incomplete (surprisingly low number of species, lack of the total numbers and/or biomass of both rotifers and crustaceans). That is why, they are unnecessary in my opinion. Moreover, these zooplankton data did not refer to bacterial and phytoplankton abundance and composition in the Discussion section (also in the Introduction); the title of the manuscript does not include zooplankton as well.
Response:
Thank you very much for your suggestions. You are right. The results for zooplankton are unnecessary. As you suggested, the methods and results for zooplankton has been deleted in the revised manuscript.
Discussion: The whole discussion is like a general review and it is based only on the comparison of the obtained results with those of other studies. The discussion should focus on the novelty of the findings obtained in this study.
It is hard to know which groups of bacteria dominated both in the studied lakes and other freshwater ecosystems (e.g. lines 338-341: “Several previous studies have reported that the bacterial communities in the freshwater lakes were mainly dominated by Proteobacteria [3,5,22,28]. Proteobacteria was also the most dominant group in both Lake South and Lake Basomtso. In addition, previous studies have showed that the β-proteobacteria group was mainly dominated in the freshwater microbial community [29,30], which was also consistent with the results in the present study” and then line 348: “Actinobacteria are the most dominant bacterial group in freshwater ecosystems [36].”).
Response:
Thank you very much for your comments and suggestions. We agree that our original discussion is boring. As you suggested, we revised the discussion to focus on the novel finding in the present study.
In the first paragraph, our discussion focused on the finding that “The bacterial taxa were more abundant in eutrophic lake than those in oligotrophic lake, but the bacterial diversity in eutrophic lake were lower than that in oligotrophic lake.”. (Line 324-334)
“In the present study, both the nutrients (TP, TN) and temperature in eutrophic Lake South were significantly higher than those in oligotrophic Lake Basomtso. As we know, the high excessive nutrient and temperature could induce the growth and reproduction of planktonic algae and significantly increase the biomass of phytoplankton [20,21,42]. This maybe the reason why the density of phytoplankton in eutrophic South Lake was higher than that in oligotrophic Lake Basomtso. Furthermore, the growth and decomposition of algae will not only increase the pH of water, but also induce the growth of bacteria [7,9]. This may explain that some bacterial taxa were more abundant in eutrophic Lake South than those in oligotrophic Lake Basomtso. However, at the same time, the growth and decomposition of harmful algae (such as Microcystis aeruginosa) could release algal toxin, which might lead to the death of some bacterial species [37,43]. This maybe the reason why the bacterial diversity in eutrophic lake was lower than that in oligotrophic lake.”
In the second paragraph, our discussion focused on the finding that “The abundance of Actinomycetes and Chloroflexi in eutrophic lake were significant higher than those in oligotrophic lake”. (Line 335-346)
“In eutrophic lake, the abundance of Actinomycetes was significantly higher than that in oligotrophic lake. Previous studies have reported that the abundance of Actinomycetes was positively correlated with the density of phytoplankton [44,45]. In addition, Pearce et al. reported that nutrient enrichment could induce an apparent shift from the β-proteobacteria to Actinobacteria in an Antarctic freshwater lake [46]. Recently, metabolic reconstruction indicated that Actinomycetes was facultative aerobes with transporters and enzymes for use of pentoses, polyamines and dipeptides [47,48]. These results suggested that the Actinobacteria could serve as saprophytic microbes that preferred eutrophic conditions. On the other hand, in the sediment samples, our study found that Chloroflexi was more abundant in eutrophic lake than that in oligotrophic lake. Chloroflexi was a photoautotrophic microorganism that was usually associated with dechlorination and could participate in the degradation of organic matter [49-51]. These may explain the higher relative abundance of the Chloroflexi in sediments of eutrophic lakes.”
In the third paragraph, our discussion focused on the finding that “both Verrucomicrobia and Acidobacteria were more abundant in oligotrophic lake compared to eutrophic lake”. (Line 347-360)
“In oligotrophic lake, Verrucomicrobia and Acidobacteria were more abundant than those in eutrophic lake. Verrucomicrobia was an obligate or facultative anaerobic bacterium [52,53]. Previous studies have showed that Verrucomicrobia preferred to grow in relatively better habitats [52,54]. Additionally, recent studies have shown that Verrucomicrobia can degrade sulfate polysaccharides, and act as polysaccharide degraders in freshwater systems [55,56]. In this study, Verrucomicrobia also exhibited a higher abundance in the waters of oligotrophic Lake Basomtso. These results suggested that Verrucomicrobia preferred to inhabit in the oligotrophic lake. In the sediments, the abundance of Acidobacteria in oligotrophic lake was significantly higher than that in eutrophic lake. Acidobacteria could inhabit a wide variety of terrestrial and aquatic habitats and were particularly abundant in acidic soils, peatlands and mineral iron-rich environments [57-59]. Many studies indicated that the abundance of the Acidobacteria could increase when the pH value was lower than 5.5 [60,61]. In this study, the pH value in the oligotrophic lake was lower than that in the eutrophic lake, possibly resulting in higher abundance of Acidobacteria in the oligotrophic lake compared to the eutrophic lake.”
In the fourth paragraph, our discussion focused on the finding that “the bacterial abundance and diversity in the sediment were both higher than those in the water samples in the two different trophic lakes.” (Line 361-368)
“Compare analysis between the water and sediments showed that the bacterial diversity and abundance in sediments were higher than those in water. Among them, the abundance of Chloroflexi in sediments was higher than that in water. Previous studies have also reported that Chloroflexi could be highly detected in sediments of many freshwater lakes [62-64]. Chloroflexi was a photoautotrophic microorganism, usually linked to dechlorination, and could participate in the degradation of organic matters [49-51]. Interestingly, our present study also found that Chloroflexi in the sediments of eutrophic lake was more abundant than that in oligotrophic lake, which further confirmed that the Chloroflexi preferred saprophytic conditions.”
In the last paragraph, we summarized our main findings in the study. (Line 369-379)
“In summary, our results displayed that there were significant differences in the bacterial and phytoplankton communities between oligotrophic lake and eutrophic lake. Firstly, the phytoplankton density and diversity in eutrophic lake were both higher than those in the oligotrophic lake. Secondly, the bacterial abundance in the eutrophic lake was higher than that in the oligotrophic lake, but the bacterial community diversity in the oligotrophic lake was higher than that in the eutrophic lake. Thirdly, the abundance of Actinomycetes and Chloroflexi in the eutrophic lake were significantly higher than those in the oligotrophic lake; in contrast, both Verrucomicrobia and Acidobacteria were more abundant in the oligotrophic lake compared to the eutrophic lake. Finally, both the bacterial abundance and diversity in sediments were higher than those in water in the two different trophic lakes. These results will further enrich our knowledge about the phytoplankton and bacterial community structure in different trophic lakes.”
Table 1: “sp.” and “nauplii” should not be in italics.
Response:
Thank you very much for your comments. As you suggested, we have corrected the italics “sp.”. (Supplemental Table 1)
Change “emergence” to “presence”
Response:
Thank you very much for your comments. As you suggested, the Supplemental Table 2 for zooplankton has been deleted in the revised manuscript.
Change “Pseudoanabnena sp”.to “Pseudanabaena sp.”
Response:
Thank you very much for your suggestions. As you suggested, we have corrected “Pseudoanabnena sp” to “Pseudanabaena sp.”. (Supplemental Table 1)
Diatoma vulgare - accepted name is Diatoma vulgaris var. vulgaris Bory de Saint-Vincent, 1824.
Response:
Thank you very much for your comments and suggestions. As you suggested, we have corrected “Diatoma vulgare” to “Diatoma vulgaris var. vulgaris Bory de Saint-Vincent”. (Supplemental Table 1)
Change „Khawkinea acutecouato” to „Khawkinea acutecaudata”; in the previous version of the manuscript, this species was not mentioned. Moreover, in the previous version, the total number of species in Lake South was 86, but in the revised version it is 97 species – why? Please check carefully other genus/species names!
Response:
Thank you very much for your comments. Sorry for our shortcoming. In the previous version of the manuscript, we lost 11 genus/species in Supplemental Table 1. The missed genus/species have been added and highlighted in present version with yellow. In addition, as you suggested, we have carefully checked and corrected other genus/species names in Supplemental Table 1.
Reviewer 2 Report
The author's have successfully addressed my conerns. The mansucript has improved a lot in terms of presentation.
Author Response
The author's have successfully addressed my conerns. The manuscript has improved a lot in terms of presentation.
Response:
Thank you very much for your help with the paper review.
Reviewer 3 Report
I found that the authors have seriously worked on finalizing the manuscript. The manuscript has become much better.
If you want to keep the phytoplankton in the manuscript, it is necessary to correct the Latin names of the species in accordance with the modern taxonomy. For example, following the database: https://www.algaebase.org.
In particular:
"Cyanophyta" better replaced by "Cyanobacteria"
genus "Anabaena" renamed to "Dolichospermum"
genus "Melosira" renamed to "Aulacoseira" over 25 years ago
Synedra ulna is currently regarded as a synonym of Ulnaria ulna (Nitzsch) Compère
and many others...
I strongly recommend that authors check the taxonomic status of species names used.
Author Response
I found that the authors have seriously worked on finalizing the manuscript. The manuscript has become much better.
Response:
Thank you very much for your help with the paper review.
If you want to keep the phytoplankton in the manuscript, it is necessary to correct the Latin names of the species in accordance with the modern taxonomy. For example, following the database: https://www.algaebase.org.
Response:
Thank you very much fro providing us the database, we have carried out a carefully checked and corrected according to the database. (Supplemental Table 1)
"Cyanophyta" better replaced by "Cyanobacteria"
Response:
Thank you very much for your suggestion. As you suggested, we have corrected “Cyanophyta” to “Cyanobacteria”. (Supplemental Table 1)
genus "Anabaena" renamed to "Dolichospermum"
Response:
Thank you very much for your suggestion. As you suggested, we have corrected “Anabaena" to "Dolichospermum”. (Supplemental Table 1)
genus "Melosira" renamed to "Aulacoseira" over 25 years ago
Response:
Thank you very much for your comments. As you suggested, we have corrected “Melosira" to "Aulacoseira”. (Supplemental Table 1)
Synedra ulna is currently regarded as a synonym of Ulnaria ulna (Nitzsch) Compère
and many others...
Response:
Thank you very much for your comments and suggestions. As you suggested, we have corrected “Synedra ulna” to “Ulnaria ulna”. (Supplemental Table 1)
I strongly recommend that authors check the taxonomic status of species names used.
Response:
Thank you very much for your suggestions. As you suggested, we have checked and corrected the taxonomic status of species names based on the database you provided. (Supplemental Table 1)
Round 3
Reviewer 1 Report
The authors have modified the manuscript according to the comments and suggestions. I recommend this manuscript for publication in Microorganisms after minor corrections.
Minor comments:
Line 29: delete “in”
Line 34: change “which” to “and”
Line 37: change “lake were” to “lakes is”
Lines 47-48: change “In the aquatic environment, the compositions and diversities” to “In aquatic ecosystems, the composition and diversity”
Line 149: change “value” to “values”
Line 166: change “numbers” to “the number” and “were” to “was”
Line 326: change “was” to “is”
Line 329: change “On the other hand, in the sediment samples, our study found that Chloroflexi was more” to “On the other hand, our study found that in the sediment samples Chloroflexi was more”
Line 332: change “These” to “This”
Line 347: change “Compare analysis between” to “Comparative analysis of”
Lines 360: change “abundance” to “abundances”
Supplemental Table 1: delete “Bory de Saint-Vincent” - the name of describer is not given for other species.